# Is Distance Matrix Enough for Geometric Deep Learning?

**Zian Li**[1,2]**, Xiyuan Wang**[1,2]**, Yinan Huang**[1]**, Muhan Zhang**[1,*]
[1]Institute for Artificial Intelligence, Peking University
[2]School of Intelligence Science and Technology, Peking University

## Abstract

Graph Neural Networks (GNNs) are often used for tasks involving the 3D geometry of a given graph, such as molecular dynamics simulation. While incorporating Euclidean distance into Message Passing Neural Networks (referred to as Vanilla DisGNN) is a straightforward way to learn the geometry, it has been demonstrated that Vanilla DisGNN is geometrically incomplete. In this work, we first construct families of novel and symmetric geometric graphs that Vanilla DisGNN cannot distinguish even when considering all-pair distances, which greatly expands the existing counterexample families. Our counterexamples show the inherent limitation of Vanilla DisGNN to capture symmetric geometric structures. We then propose $k$-DisGNNs, which can effectively exploit the rich geometry contained in the distance matrix. We demonstrate the high expressive power of $k$-DisGNNs from three perspectives: 1. They can learn high-order geometric information that cannot be captured by Vanilla DisGNN. 2. They can unify some existing well-designed geometric models. 3. They are universal function approximators from geometric graphs to scalars (when $k \geq 2$) and vectors (when $k \geq 3$). Most importantly, we establish a connection between geometric deep learning (GDL) and traditional graph representation learning (GRL), showing that those highly expressive GNN models originally designed for GRL can also be applied to GDL with impressive performance, and that existing complicated, equivariant models are not the only solution. Experiments verify our theory. Our $k$-DisGNNs achieve many new state-of-the-art results on MD17.

## 1 Introduction

Many real-world tasks are relevant to learning the geometric structure of a given graph, such as molecular dynamics simulation, physical simulation, drug designing and protein structure prediction [Schmitz et al., 2019, Sanchez-Gonzalez et al., 2020, Jumper et al., 2021, Guo et al., 2020]. Usually in these tasks, the coordinates of nodes and their individual properties, such as atomic numbers, are given. The goal is to accurately predict both invariant properties, such as the energy of the molecule, and equivariant properties, such as the force acting on each atom. This kind of graphs is also referred to as *geometric graphs* by researchers [Bronstein et al., 2021, Han et al., 2022].

In recent years, Graph Neural Networks (GNNs) have achieved outstanding performance on such tasks, as they can learn a representation for each graph or node in an end-to-end fashion, rather than relying on handcrafted features. In such setting, a straightforward idea is to incorporate Euclidean distance, the most basic geometric feature, into Message Passing Neural Network [Gilmer et al., 2017] (we call these models Vanilla DisGNN) [Schütt et al., 2018, Kearnes et al., 2016]. While Vanilla DisGNN is simple yet powerful when considering all-pair distances (we assume all-pair

---

*Corresponding author: Muhan Zhang (muhan@pku.edu.cn).

distance is used by default to research a model's maximal expressive power), it has been proved to be geometrically incomplete [Zhang et al., 2021, Garg et al., 2020, Schütt et al., 2021, Pozdnyakov and Ceriotti, 2022], i.e., there exist pairs of geometric graphs which Vanilla DisGNN cannot distinguish. It has led to the development of various GNNs which go beyond simply using *distance* as input. Instead, these models use complex group irreducible representations, first-order equivariant representations or manually-designed complex invariant geometric features such as angles or dihedral angles to learn better representations from geometric graphs. It is generally believed that pure distance information is insufficient for complex GDL tasks [Klicpera et al., 2020a, Schütt et al., 2021].

On the other hand, it is well known that the *distance matrix*, which contains the distances between all pairs of nodes in a geometric graph, holds all of the geometric structure information [Satorras et al., 2021]. This suggests that it is possible to obtain all of the desired geometric structure information from *distance graphs*, i.e., complete weighted graphs with Euclidean distance as edge weight. Therefore, the complex GDL task with node coordinates as input (i.e., 3D graph) may be transformed into an equivalent graph representation learning task with distance graphs as input (i.e., 2D graph).

Thus, a desired question is: Can we design theoretically and experimentally powerful geometric models, which can learn the rich geometric patterns purely from distance matrix?

In this work, we first revisit the counterexamples in previous work used to show the incompleteness of Vanilla DisGNN. These existing counterexamples either consider only finite pairs of distance (a cutoff distance is used to remove long edges) [Schütt et al., 2021, Zhang et al., 2021, Garg et al., 2020], which can limit the representation power of Vanilla DisGNN, or lack diversity thus may not effectively reveal the inherent limitation of Vanilla DisGNN [Pozdnyakov and Ceriotti, 2022]. In this regard, we further constructed plenty of novel and symmetric counterexamples, as well as a novel method to construct families of counterexamples based on several basic units. These counterexamples significantly enrich the current set of counterexample families, and can reveal the inherent limitations of MPNNs in *capturing symmetric configurations*, which can explain the reason why they perform badly in real-world tasks where lots of symmetric (sub-)structures are included.

Given the limitations of Vanilla DisGNN, we propose $k$-DisGNNs, models that take pair-wise distance as input and aim for learning the rich geometric information contained in geometric graphs. $k$-DisGNNs are mainly based on the well known $k$-(F)WL test [Cai et al., 1992], and include three versions: $k$-DisGNN, $k$-F-DisGNN and $k$-E-DisGNN. We demonstrate the superior geometric learning ability of $k$-DisGNNs from three perspectives. We first show that $k$-DisGNNs can capture arbitrary $k$- or $(k+1)$-order **geometric information** (multi-node features), which cannot be achieved by Vanilla DisGNN. Then, we demonstrate the high **generality** of our framework by showing that $k$-DisGNNs can implement DimeNet [Klicpera et al., 2020a] and GemNet [Gasteiger et al., 2021], two well-designed state-of-the-art models. A key insight is that these two models are both augmented with manually-designed high-order geometric features, including angles (three-node features) and dihedral angles (four-node features), which correspond to the $k$-tuples in $k$-DisGNNs. However, $k$-DisGNNs can learn *more than* these handcrafted features in an end-to-end fashion. Finally, we demonstrate that $k$-DisGNNs can act as **universal function approximators** from geometric graphs to **scalars** (i.e., E(3) invariant properties) when $k \geq 2$ and **vectors** (i.e., first-order O(3) equivariant and translation invariant properties) when $k \geq 3$. This essentially answers the question posed in our title: *distance matrix is sufficient for GDL*. We conduct experiments on benchmark datasets where our models achieve **state-of-the-art results** on a wide range of the targets in the MD17 dataset.

Our method reveals the high potential of the most basic geometric feature, *distance*. Highly expressive GNNs originally designed for traditional GRL can naturally leverage such information as edge weight, and achieve high theoretical and experimental performance in geometric settings. This **opens up a new door for GDL research** by transferring knowledge from traditional GRL, and suggests that existing complex, equivariant models may not be the only solution.

## 2 Related Work

**Equivariant neural networks.** Symmetry is a rather important design principle for GDL [Bronstein et al., 2021]. In the context of geometric graphs, it is desirable for models to be equivariant or invariant under the group (such as E(3) and SO(3)) actions of the input graphs. These models include those using group irreducible representations [Thomas et al., 2018, Batzner et al., 2022, Anderson et al., 2019, Fuchs et al., 2020], complex invariant geometric features [Schütt et al., 2018, Klicpera

et al., 2020a, Gasteiger et al., 2021] and first-order equivariant representations [Satorras et al., 2021, Schütt et al., 2021, Thölke and De Fabritiis, 2021]. Particularly, Dym and Maron [2020] proved that both Thomas et al. [2018] and Fuchs et al. [2020] are universal for SO(3)-equivariant functions. Besides, Villar et al. [2021] showed that one could construct powerful (first-order) equivariant outputs by leveraging invariances, highlighting the potential of invariant models to learn equivariant targets.

**GNNs with distance.** In geometric settings, incorporating 3D distance between nodes as edge weight is a simple yet efficient way to improve geometric learning. Previous work [Maron et al., 2019, Morris et al., 2019, Zhang and Li, 2021, Zhao et al., 2022] mostly treat distance as an auxiliary edge feature for better experimental performance and do not explore the expressiveness or performance of using pure distance for geometric learning. Schütt et al. [2018] proposes to expand distance using a radial basis function as a continuous-filter and perform convolutions on the geometric graph, which can be essentially unified into the Vanilla DisGNN category. Zhang et al. [2021], Garg et al. [2020], Schütt et al. [2021], Pozdnyakov and Ceriotti [2022] demonstrated the limitation of Vanilla DisGNN by constructing pairs of non-congruent geometric graphs which cannot be distinguished by it, thus explaining the poor performance of such models [Schütt et al., 2018, Kearnes et al., 2016]. However, none of these studies proposed a complete purely distance-based geometric model. Recent work [Hordan et al., 2023] proposed theoretically complete GNNs for distinguishing geometric graphs, but they go beyond pair-wise distance and instead utilize gram matrices or coordinate projection.

**Expressive GNNs.** In traditional GRL (no 3D information), it has been proven that the expressiveness of MPNNs is limited by the Weisfeiler-Leman test [Weisfeiler and Leman, 1968, Xu et al., 2018, Morris et al., 2019], a classical algorithm for graph isomorphism test. While MPNNs can distinguish most graphs [Babai and Kucera, 1979], they are unable to count rings, triangles, or distinguish regular graphs, which are common in real-world data such as molecules [Huang et al., 2023]. To increase the expressiveness and design space of GNNs, Morris et al. [2019, 2020] proposed high-order GNNs based on $k$-WL. Maron et al. [2019] designed GNNs based on the folklore WL (FWL) test [Cai et al., 1992] and Azizian and Lelarge [2020] showed that these GNNs are the most powerful GNNs for a given tensor order. Beyond these, there are subgraph GNNs [Zhang and Li, 2021, Bevilacqua et al., 2021, Frasca et al., 2022, Zhao et al., 2021], substructure-based GNNs [Bouritsas et al., 2022, Horn et al., 2021, Bodnar et al., 2021] and so on, which are also strictly more expressive than MPNNs. For a comprehensive review on expressive GNNs, we refer the readers to Zhang et al. [2023].

More related work can be referred to Appendix E.

## 3 Preliminaries

In this paper, we denote multiset with $\{\!\{\ \}\!\}$. We use $[n]$ to represent the set $\{1, 2, ..., n\}$. A complete weighted graph with $n$ nodes is denoted by $G = (V, \boldsymbol{E})$, where $V = [n]$ and $\boldsymbol{E} = [e_{ij}]_{n \times n} \in \mathbb{R}^{n \times n}$. The neighborhoods of node $i$ are denoted by $N(i)$.

**Distance graph vs. geometric graph.** In many tasks, we need to deal with geometric graphs, where each node $i$ is attached with its 3D coordinates $\mathbf{x}_i \in \mathbb{R}^3$ in addition to other invariant features. Geometric graphs contain rich geometric information useful for learning chemical or physical properties. However, due to the significant variation of coordinates under E(3) transformations, they can be redundant when it comes to capturing geometric structure information.

Corresponding to geometric graphs are distance graphs, i.e., *complete* weighted graph with Euclidean distance as edge weight. Unlike geometric graphs, distance graphs do not have explicit coordinates attached to each node, but instead, they possess distance features that are naturally *invariant* under E(3) transformation and can be *readily utilized* by most GNNs originally designed for traditional GRL. Distance also provides an inherent *inductive bias* for effectively modeling the interaction/relationship between nodes. Moreover, a distance graph maintains all the essential *geometric structure information*, as stated in the following theorem:

**Theorem 3.1.** *[Satorras et al., 2021] Two geometric graphs are congruent (i.e., they are equivalent by permutation of nodes and E(3) transformation of coordinates) $\iff$ their corresponding distance graphs are isomorphic.*

By additionally incorporating *orientation* information, we can learn equivariant features as well [Villar et al., 2021]. Hence, distance graphs provide a way to **represent geometric graphs without referring to a canonical coordinate system**. This urges us to study their full potential for GDL.

**Weisfeiler-Leman Algorithms.** Weisfeiler-Lehman test (also called as 1-WL) [Weisfeiler and Leman, 1968] is a well-known efficient algorithm for graph isomorphism test (traditional setting, no 3D information). It iteratively updates the labels of nodes according to nodes' own labels and their neighbors' labels, and compares the histograms of the labels to distinguish two graphs. Specifically, we use $l_i^t$ to denote the label of node $i$ at iteration $t$, then 1-WL updates the node label by

$$l_i^{t+1} = \text{HASH}(l_i^t, \{\!\{l_j^t \mid j \in N(i)\}\!\}), \tag{1}$$

where HASH is an injective function that maps different inputs to different labels. However, 1-WL cannot distinguish all the graphs [Zhang and Li, 2021], and thus $k$-dimensional WL ($k$-WL) and $k$-dimensional Folklore WL ($k$-FWL), $k \geq 2$, are proposed to boost the expressiveness of WL test.

Instead of updating the label of nodes, $k$-WL and $k$-FWL update the label of $k$-tuples $\boldsymbol{v} := (v_1, v_2, ..., v_k) \in V^k$, denoted by $l_{\boldsymbol{v}}$. Both methods initialize $k$-tuples' labels according to their isomorphic types and update the labels iteratively according to their $j$-neighbors $N_j(\boldsymbol{v})$. The difference between $k$-WL and $k$-FWL mainly lies in the definition of $N_j(\boldsymbol{v})$. To be specific, tuple $\boldsymbol{v}$'s $j$-neighbors in $k$-WL and $k$-FWL are defined as follows respectively

$$N_j(\boldsymbol{v}) = \{\!\{(v_1, ..., v_{j-1}, w, v_{j+1}, ..., v_k) \mid w \in V\}\!\}, \tag{2}$$

$$N_j^F(\boldsymbol{v}) = \big((j, v_2, ..., v_k), (v_1, j, ..., v_k), ..., (v_1, v_2, ..., j)\big). \tag{3}$$

To update the label of each tuple, $k$-WL and $k$-FWL iterates as follows

$$k\text{-WL}: \quad l_{\boldsymbol{v}}^{t+1} = \text{HASH}\Big(l_{\boldsymbol{v}}^t, \big(\{\!\{l_{\boldsymbol{w}}^t \mid \boldsymbol{w} \in N_j(\boldsymbol{v})\}\!\} \mid j \in [k]\big)\Big), \tag{4}$$

$$k\text{-FWL}: \quad l_{\boldsymbol{v}}^{t+1} = \text{HASH}\Big(l_{\boldsymbol{v}}^t, \{\!\{\big(l_{\boldsymbol{w}}^t \mid \boldsymbol{w} \in N_j^F(\boldsymbol{v})\big) \mid j \in [|V|]\}\!\}\Big). \tag{5}$$

According to Cai et al. [1992], there always exists a pair of non-isomorphic graphs that cannot be distinguished by $k$-(F)WL but can be distinguished by $(k+1)$-(F)WL. This means that $k$-(F)WL forms a strict hierarchy, which, however, still cannot solve the graph isomorphism problem with a finite $k$.

# 4 Revisiting the Incompleteness of Vanilla DisGNN

As mentioned in previous sections, Vanilla DisGNN is the edge-enhanced version of MPNN, which can unify many model frameworks [Schütt et al., 2018, Kearnes et al., 2016]. Its message passing formula can be generalized from its discrete version, which we call 1-WL-E [Pozdnyakov and Ceriotti, 2022]:

$$l_i^{t+1} = \text{HASH}(l_i^t, \{\!\{(l_j^t, e_{ij}) \mid j \in |V|\}\!\}). \tag{6}$$

To provide valid counterexamples that Vanilla DisGNN cannot distinguish, Pozdnyakov and Ceriotti [2022] proposed both finite-size and periodic counterexamples and showed that they included real chemical structures. This work demonstrated for the first time the inherent limitations of Vanilla DisGNN. However, it should be noted that the essential change between these counterexamples is merely the distortion of $C^{\pm}$ atoms, which lacks diversity despite having high manifold dimensions. In this regard, constructing new families of counterexamples cannot only enrich the diversity of existing families, but also demonstrate Vanilla DisGNN's limitations from different angles.

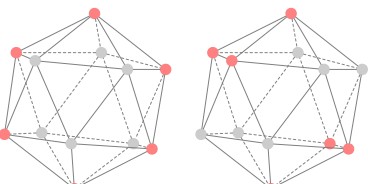

Figure 1: A pair of geometric graphs that are non-congruent but cannot be distinguished by Vanilla DisGNN. Only red nodes belong to the two geometric graphs. The grey nodes and the "edges" are for visualization purpose only. Note that the nodes of the geometric graphs are sampled from regular icosahedrons.

In this paper, we give a **simple** valid counterexample which Vanilla DisGNN cannot distinguish even with an infinite cutoff, as shown in Figure 1. In both geometric graphs, all nodes have exactly the same unordered list of distances (infinite cutoff considered), which means that Vanilla DisGNN will always label them identically. Nevertheless, the two geometric graphs are obviously non-congruent, since there are two small equilateral triangles on the right, and zero on the left. Beyond this, we construct three kinds of counterexamples in Appendix A, which can significantly enrich the counterexamples found by Pozdnyakov and Ceriotti [2022], including:

1. Individual counterexamples sampled from regular polyhedrons that can be directly verified.

2. Small families of counterexamples that can be transformed in one dimension.

3. Families of counterexamples constructed by *arbitrary combinations* of basic symmetric units.

These counterexamples further demonstrate the incompleteness of Vanilla DisGNN in learning the geometry: even quite simple geometric graphs as shown in Figure 1 cannot be distinguished by it. While the counterexamples we constructed may not correspond to real molecules, such symmetric structures are commonly seen in other geometry-relevant tasks, such as physical simulations and point clouds. The inability to distinguish these symmetric structures can have a significant impact on the geometric learning performance of models, even for non-degenerated configurations, as demonstrated by Pozdnyakov and Ceriotti [2022]. This perspective can provide an additional explanation for the poor performance of models such as SchNet [Schütt et al., 2018] and inspire the design of more powerful and efficient geometric models.

## 5   $k$-DisGNNs: $k$-order Distance Graph Neural Networks

In this section, we propose the framework of $k$-DisGNNs, *complete and universal* geometric models learning from *pure distance features*. $k$-DisGNNs consist of three versions: $k$-DisGNN, $k$-F-DisGNN, and $k$-E-DisGNN. Detailed implementation is referred to Appendix C.

**Initialization Block.** Given a geometric $k$-tuple $\boldsymbol{v}$, high-order DisGNNs initialize it with an injective function by $h_{\boldsymbol{v}}^0 = f_{\text{init}}\Big( \big( z_{v_i} \mid i \in [k] \big), \big( e_{v_i v_j} \mid i, j \in [k], i < j \big) \Big) \in \mathbb{R}^K$, where $K$ is the hidden dimension. This function can injectively embed the *ordered* distance matrix along with the $z$ type of each node within the $k$-tuple, thus preserving all the geometric information within it.

**Message Passing Block.** The key difference between the three versions of high-order DisGNNs lies in their message passing blocks. The message passing blocks of $k$-DisGNN and $k$-F-DisGNN are based on the paradigms of $k$-WL and $k$-FWL, respectively. Their core message passing function $f_{\text{MP}}^t$ and $f_{\text{MP}}^{\text{F},t}$ are formulated simply by replacing the *discrete* tuple labels $l_{\boldsymbol{v}}^t$ in $k$-(F)WL (Equation (4, 5)) with *continuous* geometric tuple representation $h_{\boldsymbol{v}}^t \in \mathbb{R}^K$, see Equation (7, 8). Tuples containing local geometric information interact with each other in message passing blocks, thus allowing models to learn considerable global geometric information (as explained in Section 6.1).

$$k-\text{DisGNN}: h_{\boldsymbol{v}}^{t+1} = f_{\text{MP}}^t \Big( h_{\boldsymbol{v}}^t, \big( \{\!\{ h_{\boldsymbol{w}}^t \mid \boldsymbol{w} \in N_j(\boldsymbol{v}) \}\!\} \mid j \in |k| \big) \Big), \tag{7}$$

$$k-\text{F}-\text{DisGNN}: h_{\boldsymbol{v}}^{t+1} = f_{\text{MP}}^{\text{F},t} \Big( h_{\boldsymbol{v}}^t, \{\!\{ \big( h_{\boldsymbol{w}}^t \mid \boldsymbol{w} \in N_j^F(\boldsymbol{v}) \big) \mid j \in [|V|] \}\!\} \Big). \tag{8}$$

However, during message passing in $k$-DisGNN, the information about distance is not explicitly used but is embedded implicitly in the initialization block when each geometric tuple is embedded according to its distance matrix and node types. This means that $k$-DisGNN is **unable to capture the relationship** between a $k$-tuple $\boldsymbol{v}$ and its neighbor $\boldsymbol{w}$ during message passing, which could be very helpful for learning the geometric structural information of the graph. For example, in a physical system, a tuple $\boldsymbol{w}$ far from $\boldsymbol{v}$ should have much less influence on $\boldsymbol{v}$ than another tuple $\boldsymbol{w}'$ near $\boldsymbol{v}$.

Based on this observation, we propose $k$-**E-DisGNN**, which maintains a representation $e_{ij}^t$ for each edge $e_{ij}$ at every time step and explicitly incorporates it into the message passing procedure. The edge representation $e_{ij}^t$ is defined as

$$e_{ij}^t = f_{\text{e}}^t \Big( e_{ij}, \big( \{\!\{ h_{\boldsymbol{w}}^t \mid \boldsymbol{w} \in V^k, \boldsymbol{w}_u = i, \boldsymbol{w}_v = j \}\!\} \mid u, v \in [k], u < v \big) \Big). \tag{9}$$

Note that $e_{ij}^t$ not only contains the distance between nodes $i$ and $j$, but also pools all the tuples related to edge $ij$, making it informative and general. For example, in the special case $k = 2$, Equation (9) is equivalent to $e_{ij}^t = f_{\text{e}}^t(e_{ij}, h_{ij}^t)$.

We realize the message passing function of $k$-E-DisGNN, $f_{\text{MP}}^{\text{E},t}$, by replacing the neighbor representation, $h_{\boldsymbol{w}}^t$ in $f_{\text{MP}}^t$, with $\big( h_{\boldsymbol{w}}^t, e_{\boldsymbol{v} \backslash \boldsymbol{w}, \boldsymbol{w} \backslash \boldsymbol{v}}^t \big)$ where $\boldsymbol{v} \backslash \boldsymbol{w}$ gives the only element in $\boldsymbol{v}$ but not in $\boldsymbol{w}$, see Equation 10. In other words, $e_{\boldsymbol{v} \backslash \boldsymbol{w}, \boldsymbol{w} \backslash \boldsymbol{v}}^t$ gives the representation of the edge connecting $\boldsymbol{v}, \boldsymbol{w}$. By this means, a tuple can be aware of **how far** it is to its neighbors and by what kind of edges each neighbor

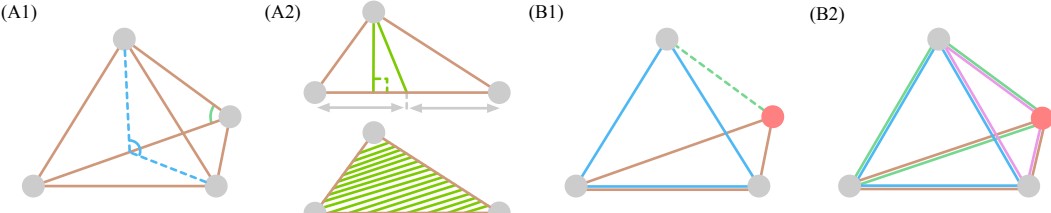

Figure 2: **A**: High-order geometric information contained in the distance matrix of $k$-tuples. We mark different orders with different colors, with brown, green, blue for 2-,3-,4-order respectively. (A1) High-order geometric information contained in 4-tuples, including distances, angles and dihedral angles. (A2) More 3-order geometric features, such as vertical lines, middle lines and the area of triangles. **B**: Examples explaining that neighboring 3-tuples can form a 4-tuple. Blue represents the center tuple, the other colors represent neighbor tuples, and the red node is the one node of that neighbor tuple which is not in the center tuple. (B1) Example for 3-E-DisGNN. With the green edge, two 3-tuples can form a 4-tuple. (B2) Example for 3-F-DisGNN. Four 3-tuples form a 4-tuple.

is connected. This can boost the ability of geometric structure learning (as explained in Section 6.1).

$$k-\text{E} - \text{DisGNN} : h_{\boldsymbol{v}}^{t+1} = f_{\text{MP}}^{\text{E},t}\Big(h_{\boldsymbol{v}}^t, \big(\{\!\!\{(h_{\boldsymbol{w}}^t, e_{\boldsymbol{v}\backslash\boldsymbol{w},\boldsymbol{w}\backslash\boldsymbol{v}}^t) \mid \boldsymbol{w} \in N_j(\boldsymbol{v})\}\!\!\} \mid j \in |k|\big)\Big). \tag{10}$$

**Output Block.** The output function $t = f_{\text{out}}\big(\{\!\!\{h_{\boldsymbol{v}}^T \mid \boldsymbol{v} \in V^k\}\!\!\}\big)$, where $T$ is the final iteration, injectively pools all the tuple representations, and generates the E(3) and permutation invariant geometric target $t \in \mathbb{R}$.

Like the conclusion about $k$-(F)WL for unweighted graphs, the expressiveness (in terms of approximating functions) of DisGNNs does *not decrease* as $k$ increases. This is simply because all $k$-tuples are contained within some $(k + 1)$-tuples, and by designing message passing that ignores the last index, we can implement $k$-DisGNNs with $(k + 1)$-DisGNNs.

## 6   Rich Geometric Information Learned by $k$-DisGNNs

In this section, we aim to delve deeper into the geometry learning capability of $k$-DisGNNs from various perspectives. In Subsection 6.1, we will examine the **high-order geometric features** that the models can extract from geometric graphs. This analysis is more intuitive and aids in comprehending the high geometric expressiveness of $k$-DisGNNs. In Subsection 6.2, we will show that DimeNet and GemNet, two classical and widely-used GNNs for GDL employing invariant geometric representations, are **special cases** of $k$-DisGNNs, highlighting the generality of $k$-DisGNNs. In the final subsection, we will show the **completeness** (distinguishing geometric graphs) and **universality** (approximating functions over geometric graphs) of $k$-DisGNNs, thus answering the question posed in the title: distance matrix is enough for GDL.

### 6.1   Ability of Learning High-Order Geometry

We first give the concept of *high-order geometric information* for better understanding.

**Definition 6.1.** $k$-order geometric information is the *E(3)-invariant* features calculable from $k$ nodes' 3D coordinates.

For example, in a 4-tuple, one can find various high-order geometric information, including distance (2-order), angles (3-order) and dihedral angles (4-order), as shown in Figure 2(A1). Note that high-order information is not limited to the common features listed above: we show some of other possible 3-order geometric features in Figure 2(A2).

We first note that $k$-DisGNNs can learn and process at least $k$-order geometric information. Theorem 3.1 states that we can reconstruct the whole geometry of the $k$-tuple from the embedding of its distance matrix. In other words, we can extract all the desired $k$-order geometric information solely from the embedding of the $k$-tuple's distance matrix, which is calculated at the initialization step of $k$-DisGNNs, with a learnable function.

Furthermore, both $k$-E-DisGNN and $k$-F-DisGNN can actually **learn $(k + 1)$-order geometric information** in their message passing layers. For $k$-E-DisGNN, information about $(h_{\boldsymbol{v}}, h_{\boldsymbol{w}}, e_{\boldsymbol{v} \backslash \boldsymbol{w}, \boldsymbol{w} \backslash \boldsymbol{v}})$, where $\boldsymbol{w}$ is some $j$-neighbor of $\boldsymbol{v}$, is included in the input of its update function. Since the distance matrices of tuples $\boldsymbol{v}$ and $\boldsymbol{w}$ can be reconstructed from $h_{\boldsymbol{v}}$ and $h_{\boldsymbol{w}}$, the all-pair distances of $(\boldsymbol{v}_1, \boldsymbol{v}_2, ..., \boldsymbol{v}_k, \boldsymbol{w} \backslash \boldsymbol{v})$ can be reconstructed from $(h_{\boldsymbol{v}}, h_{\boldsymbol{w}}, e_{\boldsymbol{v} \backslash \boldsymbol{w}, \boldsymbol{w} \backslash \boldsymbol{v}})$, as shown in Figure 2(B1). Similarly, the distance matrix of $(\boldsymbol{v}_1, \boldsymbol{v}_2, ..., \boldsymbol{v}_k, j)$ can also be reconstructed from $\left(h_{\boldsymbol{v}}, (h_{\boldsymbol{w}}^t \mid \boldsymbol{w} \in N_j^F(\boldsymbol{v}))\right)$ in update function of $k$-F-DisGNN as shown in Figure 2(B2). These enable $k$-E-DisGNN and $k$-F-DisGNN to reconstruct all $(k + 1)$-tuples' distance matrices during message passing and thus learn all the $(k + 1)$-order geometric information contained in the graph.

With the ability to learn high-order geometric information, $k$-DisGNNs can learn geometric structures that cannot be captured by Vanilla DisGNNs. For example, consider the counterexample shown in Figure 1. The two geometric graphs have a different number of small equilateral triangles, which cannot be distinguished by Vanilla DisGNNs even with infinite message passing layers. However, 3-DisGNN and 2-E/F-DisGNN can easily distinguish the graphs by counting the number of small equilateral triangles, which is actually a kind of 3-order geometric information. This example also illustrates that high-order DisGNNs are *strictly more powerful* than Vanilla DisGNNs.

## 6.2 Unifying Existing Geometric Models with DisGNNs

There have been many attempts to improve GDL models by *manually* designing and incorporating high-order geometric features such as angles (3-order) and dihedral angles (4-order). These features are all invariant geometric features that can be learned by some $k$-DisGNNs. It is therefore natural to ask whether these models can be implemented by $k$-DisGNNs. In this subsection, we show that two classical models, DimeNet [Klicpera et al., 2020a] and GemNet [Gasteiger et al., 2021], are just special cases of $k$-DisGNNs, thus unifying existing methods based on *hand-crafted* features with a learning paradigm that learns *arbitrary* high-order features from distance matrix.

DimeNet embeds atom $i$ with a set of incoming messages $m_{ji}$, i.e., $h_i = \sum_{j \in N_i} m_{ji}$, and updates the message $m_{ji}$ by

$$m_{ji}^{t+1} = f_{\mathrm{MP}}^{\mathrm{D}}\big(m_{ji}^t, \sum_k f_{\mathrm{int}}^{\mathrm{D}}(m_{kj}^t, d_{ji}, \phi_{kji})\big), \tag{11}$$

where $d_{ji}$ is the distance between node $j$ and node $i$, and $\phi_{kji}$ is the angle $kji$. We simplify the subscript of $\sum$, same for that in Equation (12). The detailed ones are referred to Appendix B.2, B.1.

DimeNet is one early approach that uses geometric features to improve the geometry learning ability of GNNs, especially useful for learning the *angle-relevant* energy or other chemical properties. Note that the angle information that DimeNet explicitly incorporates into its message passing function is actually a kind of 3-order geometric information, which can be exploited from distance matrices by our 2-E/F-DisGNN with a learning paradigm. This gives the key insight for the following proposition.

**Proposition 6.2.** *2-E-DisGNN and 2-F-DisGNN can implement DimeNet.*

GemNet is also a graph neural network designed to process graphs embedded in Euclidean space. Unlike DimeNet, GemNet incorporates both angle and *dihedral angle* information into the message passing functions, with a 2-step message passing scheme. The core procedure is as follows:

$$m_{ca}^{t+1} = f_{\mathrm{MP}}^{\mathrm{G}}\big(m_{ca}^t, \sum_{b,d} f_{\mathrm{int}}^{\mathrm{G}}(m_{db}^t, d_{db}, \phi_{cab}, \phi_{abd}, \theta_{cabd})\big), \tag{12}$$

where $\theta_{cabd}$ is the dihedral angle of planes $cab$ and $abd$. The use of dihedral angle information allows GemNet to learn at least 4-order geometric information, making it much more powerful than models that only consider angle information. We now prove that GemNet is just a special case of 3-E/F-DisGNN, which can also learn 4-order geometric information during its message passing stage.

**Proposition 6.3.** *3-E-DisGNN and 3-F-DisGNN can implement GemNet.*

It is worth noting that while both $k$-DisGNNs and existing models can learn some $k$-order geometric information, $k$-DisGNNs have the advantage of **learning arbitrary $k$-order geometric information**. This means that we can learn different high-order geometric features according to specific tasks in a *data-driven* manner, including but not limited to angles and dihedral angles.

## 6.3 Completeness and Universality of $k$-DisGNNs

We have provided an illustrative example in Section 6.1 that demonstrates the ability of $k$-DisGNNs to distinguish certain geometric graphs, which is not achievable by Vanilla DisGNNs. Actually, all counterexamples presented in Appendix A can be distinguished by 3-DisGNNs or 2-E/F-DisGNNs. This leads to a natural question: Can $k$-DisGNNs distinguish all geometric graphs with a finite and small value of $k$? Furthermore, can they approximate arbitrary functions over geometric graphs?

We now present our main findings that elaborate on the **completeness** and **universality** of $k$-DisGNNs with finite and small $k$, which essentially highlight the high theoretical expressiveness of $k$-DisGNNs.

**Theorem 6.4.** *(informal) Let $\mathcal{M}(\theta) \in \mathrm{MinMods} = \{$1-round 4-DisGNN, 2-round 3-E/F-DisGNN$\}$ with parameters $\theta$. Denote $h_m = f_{\mathrm{node}}\big(\{\!\{h_{\boldsymbol{v}}^T \mid \boldsymbol{v}_0 = m\}\!\}\big) \in \mathbb{R}^{K'}$ as node $m$'s representation produced by $\mathcal{M}$ where $f_{\mathrm{node}}$ is an injective multiset function, and $\mathbf{x}_m^c$ as node $m$'s coordinates w.r.t. the center. Denote $\mathrm{MLPs} : \mathbb{R}^{K'} \to \mathbb{R}$ as a multi-layer perceptron. Then we have:*

1. *(**Completeness**) There exists $\theta_0$ such that $\mathcal{M}(\theta_0)$ can distinguish all pairs of non-congruent geometric graphs.*

2. *(**Universality for scalars**) $\mathcal{M}(\theta)$ is a universal function approximator for continuous, $E(3)$ and permutation invariant functions $f : \mathbb{R}^{3 \times n} \to \mathbb{R}$ over geometric graphs.*

3. *(**Universality for vectors**) $f_{\mathrm{out}}^{\mathrm{equiv}} = \sum_{m=1}^{|V|} \mathrm{MLPs}\big(h_m\big)\mathbf{x}_m^c$ is a universal function approximator for continuous, $O(3)$ equivariant and translation and permutation invariant functions $f : \mathbb{R}^{3 \times n} \to \mathbb{R}^3$ over geometric graphs.*

In fact, completeness directly implies universality for scalars [Hordan et al., 2023]. Since we can actually recover the whole geometry from arbitrary node representation $h_m$, universality for vectors can be proved with conclusions from Villar et al. [2021]. Also note that as order $k$ and round number go higher, the completeness and universality still hold. Formal theorems and detailed proof are provided in Appendix B.3.

There is a concurrent work by Delle Rose et al. [2023], which also investigates the completeness of $k$-WL on distance graphs. In contrast to our approach of using only one tuple's final representation to reconstruct the geometry, they leverage all tuples' final representations. They prove that with the distance matrix, 3-round geometric 2-FWL and 1-round geometric 3-FWL are complete for geometry. This conclusion can also be extended to $k$-DisGNNs, leading to the following theorem:

**Theorem 6.5.** *The completeness and universality for scalars stated in Theorem 6.4 hold for $\mathcal{M}(\theta) \in \mathrm{MinMods}_{\mathrm{ext}} = \{$3-round 2-E/F-DisGNN, 1-round 3-E/F-DisGNN$\}$ with parameters $\theta$.*

This essentially lowers the required order $k$ for achieving universality on scalars to 2. However, due to the lower order and fewer rounds, the expressiveness of a single node's representation may be limited, thus the universality for vectors stated in Theorems 6.4 may not be guaranteed.

# 7 Experiments

In this section, we evaluate the experimental performance of $k$-DisGNNs. Our main objectives are to answer the following questions:

**Q1** Does 2/3-E/F-DisGNN outperform their counterparts in experiments (corresponding to Section 6.2)?

**Q2** As universal models for scalars (when $k \geq 2$), do $k$-DisGNNs also have good experimental performance (corresponding to Section 6.3)?

**Q3** Does incorporating well-designed edge representations in the $k$-E-DisGNN result in improved performance?

The best and the second best results are shown in **bold** and underline respectively in tables. Detailed experiment configuration and supplementary experiment information can be found in Appendix D. Our code is available at `https://github.com/GraphPKU/DisGNN`.

**MD17.** MD17 [Chmiela et al., 2017] is a dataset commonly used to evaluate the performance of machine learning models in the field of molecular dynamics. It contains a collection of molecular

Table 1: MAE loss on MD17. Energy (E) in kcal/mol, force (F) in kcal/mol/Å. We color the cell if DisGNNs outperforms its counterpart. The improvement ratio is calculated relative to DimeNet. $k$-DisGNNs rank **top 2** on force prediction (which determines the accuracy of molecular dynamics [Gasteiger et al., 2021]) on average.

| Target | | FCHL | PaiNN | NequIP | TorchMD | GNN-LF | DimeNet | 2F-Dis. | GemNet | 3E-Dis. |
|---|---|---|---|---|---|---|---|---|---|---|
| aspirin | E | 0.182 | 0.167 | - | **0.124** | 0.1342 | 0.204 | 0.1305 | - | 0.1466 |
| | F | 0.478 | 0.338 | 0.348 | 0.255 | 0.2018 | 0.499 | **0.1393** | 0.2168 | 0.2060 |
| benzene | E | - | - | - | **0.056** | 0.0686 | 0.078 | 0.0683 | - | 0.0795 |
| | F | - | - | 0.187 | 0.201 | 0.1506 | 0.187 | 0.1474 | **0.1453** | 0.1471 |
| ethanol | E | 0.054 | 0.064 | - | 0.054 | 0.0520 | 0.064 | **0.0502** | - | 0.0541 |
| | F | 0.136 | 0.224 | 0.208 | 0.116 | 0.0814 | 0.230 | **0.0478** | 0.0853 | 0.0617 |
| malonal. | E | 0.081 | 0.091 | - | 0.079 | 0.0764 | 0.104 | **0.0730** | - | 0.0739 |
| | F | 0.245 | 0.319 | 0.337 | 0.176 | 0.1259 | 0.383 | **0.0786** | 0.1545 | 0.0974 |
| napthal. | E | 0.117 | 0.166 | - | **0.085** | 0.1136 | 0.122 | 0.1146 | - | 0.1135 |
| | F | 0.151 | 0.077 | 0.097 | 0.06 | 0.0550 | 0.215 | 0.0518 | 0.0553 | **0.0478** |
| salicyl. | E | 0.114 | 0.166 | - | **0.094** | 0.1081 | 0.134 | 0.1071 | - | 0.1084 |
| | F | 0.221 | 0.195 | 0.238 | 0.135 | 0.1005 | 0.374 | **0.0862** | 0.1048 | 0.1186 |
| toluene | E | 0.098 | 0.095 | - | **0.074** | 0.0930 | 0.102 | 0.0922 | - | 0.1004 |
| | F | 0.203 | 0.094 | 0.101 | 0.066 | 0.0543 | 0.216 | **0.0395** | 0.0600 | 0.0455 |
| uracil | E | 0.104 | 0.106 | - | **0.096** | 0.1037 | 0.115 | 0.1036 | - | 0.1037 |
| | F | 0.105 | 0.139 | 0.173 | 0.094 | **0.0751** | 0.301 | 0.0876 | 0.0969 | 0.0921 |
| Avg improv. | E | 11.58% | -2.09% | - | **26.40%** | 17.05% | 0.00% | 18.18% | - | 13.51% |
| Rank | | 5 | 7 | - | 1 | 3 | 6 | 2 | - | 4 |
| Avg improv. | F | 31.85% | 39.13% | 29.86% | 52.40% | 63.53% | 0.00% | **69.68%** | 60.96% | 65.27% |
| Rank | | 7 | 6 | 8 | 5 | 3 | 9 | 1 | 4 | 2 |

dynamics simulations of small organic molecules such as aspirin and ethanol. Given the atomic numbers and coordinates, the task is to predict the energy of the molecule and the atomic forces. We mainly focus on the comparison between 2-F-DisGNN/3-E-DisGNN and DimeNet/GemNet. At the same time, we also compare $k$-DisGNNs with the state-of-the-art models: FCHL [Christensen et al., 2020], PaiNN [Schütt et al., 2021], NequIP [Batzner et al., 2022], TorchMD [Thölke and De Fabritiis, 2021], GNN-LF [Wang and Zhang, 2022]. The results are shown in Table 1. 2-F-DisGNN and 3-E-DisGNN outperform their counterparts on **16/16** and 6/8 targets, respectively, with an average improvement of 43.9% and 12.23%, suggesting that data-driven models can subsume carefully designed manual features given high expressiveness. In addition, $k$-DisGNNs also achieve the **best performance** on 8/16 targets, and achieve the second-best performance on **all the other targets**, outperforming the best baselines GNN-LF and TorchMD by 10.19% and 12.32%, respectively. Note that GNN-LF and TorchMD are all complex equivariant models. Beyond these, we perform experiments on **revised MD17**, which has better data quality, and $k$-DisGNNs outperforms the SOTA models [Batatia et al., 2022, Musaelian et al., 2023] on a wide range of targets. The results are referred to Appendix D.2. The results firmly answer **Q1** and **Q2** posed at the beginning of this section and demonstrate the potential of pure distance-based methods for graph deep learning.

**QM9.** QM9 [Ramakrishnan et al., 2014, Wu et al., 2018] consists of 134k stable small organic molecules with 19 regression targets. The task is like that in MD17, but this time we want to predict the molecule's properties such as the dipole moment. We mainly compare 2-F-DisGNN with DimeNet on this dataset and the results are shown in Table 2. 2-F-DisGNN outperforms DimeNet on **11/12** targets, by 14.27% on average, especially on targets $\mu$ (65.0%) and $\langle R^2 \rangle$ (90.4%), which also answers **Q1** well. A full comparison to other state-of-the-art models is included in Appendix D.2.

**Effectiveness of edge representations.** To answer **Q3**, we split the edge representation $e_{ij}^t$ (Equation (9)) into two parts, namely the pure distance (the first element) and the tuple representations (the second element), and explore whether incorporating the two elements is beneficial. We conduct experiments on MD17 with three versions of 2-DisGNNs, including 2-DisGNN (no edge representation), 2-e-DisGNN (add only edge weight) and 2-E-DisGNN (add full edge representation). The results are shown in Table 3. Both 2-e-DisGNN and 2-E-DisGNN exhibit significant performance improvements over 2-DisGNN, highlighting the significance of edge representation. Furthermore, 2-E-DisGNN outperforms 2-DisGNN and 2-e-DisGNN on 16/16 and 12/16 targets, respectively, with average improvements of 39.8% and 3.8%. This verifies our theory that capturing the *full edge representation* connecting two tuples can boost the model's representation power.

Table 2: Comparison of 2-F-DisGNN and DimeNet on QM9.

| Target | Unit | DimeNet | 2F-Dis. |
|--------|------|---------|---------|
| $\mu$ | D | 0.0286 | **0.0100** |
| $\alpha$ | $a_0^3$ | 0.0469 | **0.0431** |
| $\epsilon_{HOMO}$ | meV | 27.8 | **21.81** |
| $\epsilon_{LUMO}$ | meV | 19.7 | 21.22 |
| $\Delta\epsilon$ | meV | 34.8 | **31.3** |
| $\langle R^2 \rangle$ | $a_0^2$ | 0.331 | **0.0299** |
| ZPVE | meV | 1.29 | **1.26** |
| $U_0$ | meV | 8.02 | **7.33** |
| $U$ | meV | 7.89 | **7.37** |
| $H$ | meV | 8.11 | **7.36** |
| $G$ | meV | 8.98 | **8.56** |
| $c_v$ | cal/mol/K | 0.0249 | **0.0233** |

Table 3: Effectiveness of edge representations on MD17. Energy (E) in kcal/mol, force (F) in kcal/mol/Å.

| Target | | 2-Dis. | 2e-Dis. | 2E-Dis. |
|--------|---|--------|---------|---------|
| aspirin | E | 0.2120 | 0.1362 | **0.1280** |
| | F | 0.4483 | 0.1749 | **0.1279** |
| benzene | E | 0.1533 | 0.0723 | **0.0700** |
| | F | 0.2049 | **0.1475** | 0.1529 |
| ethanol | E | 0.0529 | 0.0506 | **0.0502** |
| | F | 0.0850 | 0.0533 | **0.0438** |
| malonal. | E | 0.0868 | 0.0736 | **0.0732** |
| | F | 0.2124 | 0.0981 | **0.0861** |
| napthal. | E | 0.1163 | **0.1133** | 0.1149 |
| | F | 0.1318 | **0.0407** | 0.0531 |
| salicyl. | E | 0.1209 | 0.1089 | **0.1074** |
| | F | 0.2723 | 0.0960 | **0.0806** |
| toluene | E | 0.1437 | 0.0924 | **0.0920** |
| | F | 0.1229 | 0.0528 | **0.0431** |
| uracil | E | 0.1152 | 0.1041 | **0.1037** |
| | F | 0.2631 | **0.0874** | 0.0933 |

## 8 Conclusions and Limitations

**Conclusions.** In this work we have thoroughly studied the ability of GNNs to learn the geometry of a graph solely from its distance matrix. We expand on the families of counterexamples that Vanilla DisGNNs are unable to distinguish from their distance matrices by constructing families of symmetric and diverse geometric graphs, revealing the inherent limitation of Vanilla DisGNNs in capturing symmetric configurations. To better leverage the geometric structure information contained in distance graphs, we proposed $k$-DisGNNs, geometric models with high generality (ability to unify two classical and widely-used models, DimeNet and GemNet), provable completeness (ability to distinguish all pairs of non-congruent geometric graphs) and universality (ability to universally approximate scalar functions when $k \geq 2$ and vector functions when $k \geq 3$). In experiments, $k$-DisGNNs outperformed previous state-of-the-art models on a wide range of targets of the MD17 dataset. Our work reveals the potential of using expressive GNN models, which were originally designed for traditional GRL, for the GDL tasks, and opens up new opportunities for this domain.

**Limitations.** Although DisGNNs can achieve universality (for scalars) with a small order of 2, the results are still generated under the assumption that the distance graph is complete. In practice, for large geometric graphs such as proteins, this assumption may lead to intractable computation due to the $O(n^3)$ time complexity and $O(n^2)$ space complexity. However, we can still balance theoretical universality and experimental tractability by using either an appropriate cutoff, which can furthest preserve the completeness of the distance matrix in local clusters while cutting long dependencies to improve efficiency (and generalization), or using sparse but expressive GNNs such as subgraph GNNs [Zhang and Li, 2021]. We leave it for future work.

## Acknowledgement

Muhan Zhang is partially supported by the National Natural Science Foundation of China (62276003) and Alibaba Innovative Research Program.

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

# A  Counterexamples for Vanilla DisGNNs

In this section, we will give some counterexamples, i.e., pairs of geometric graphs that are not congruent but cannot be distinguished by Vanilla DisGNNs. We will organize the section as follows: First, we will give some isolated counterexamples that can be directly verified; Then, we will give some counterexamples that can hold the property (i.e., to be counterexamples) after some simple continuous transformation; Finally, we will give a family of counterexamples that can be obtained by the combination of some basic units. Since the latter two cases cannot be directly verified, we will also give the detailed proof.

We say there are *M kinds* of nodes if there are $M$ different labels after infinite iterations of Vanilla DisGNNs. Note that in this section, all the grey nodes and the "edges" are just for **visualization purpose**. Only colored nodes belong to the geometric graph.

Note that we provide **verification programs** in our code for all the counterexamples. The first type of counterexamples (Appendix A.1) is limited in quantity and can be directly verified. The remaining two types (Appendix A.2 and Appendix A.3) are families of counterexamples and have an infinite number of cases. Our code can verify them to some extent by randomly selecting some parameters. Theoretical proofs for these two types are also provided in Appendix A.2 and Appendix A.3, respectively.

## A.1  Counterexamples That Can Be Directly Verified

Since the distance graph, which DisGNNs take as input, is a kind of complete graph and contains lots of geometric constraints, if we want to get a pair of non-congruent geometric graphs that cannot be distinguished by DisGNNs, they must exhibit great *symmetry* and have just *minor difference*. Inspired by this, we can try to sample nodes from regular polyhedrons, which themselves exhibit high symmetry. The first pair of geometric graphs is sampled from regular icosahedrons, which is referred to the main body of our paper, see Figure 1. Since all the nodes from both graphs have the same unordered distance list, there is only **1** kind of nodes for both graphs. We note that it is the only counterexample we can sample from just regular icosahedron.

The following pairs of geometric graphs are all sampled from regular dodecahedrons. We will distinguish them by the graph size.

**6-size and 14-size counterexamples.**

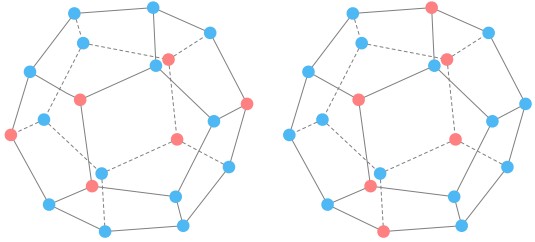

A pair of 6-size geometric graphs (red nodes) and a pair of 14-size geometric graphs (blue nodes) that are counterexamples are shown in the figure above. They are actually complementary in terms of forming all vertices of a regular dodecahedron.

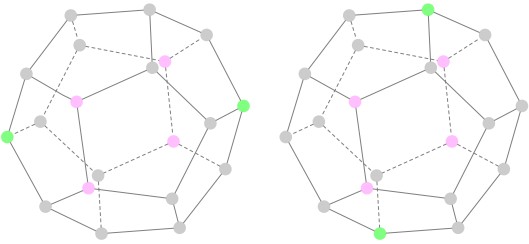

For the 6-size counterexample, the label distribution after infinite rounds of Vanilla DisGNN is shown in the picture above, and the nodes are colored differently to represent different labels. It can be directly verified that there are **2** kinds of nodes. Note that the two geometric graphs are non-congruent: The isosceles triangle of the two geometric graphs formed by one green point and two pink nodes have the same leg length but different base lengths.

For the 14-size counterexample, we give the conclusion that there are **4** kinds of nodes and the number of nodes of each kind is 2, 4, 4, 4 respectively. Since the counterexamples here and in the following are complex and not so intuitionistic, we recommend readers to directly verify them with our programs.

**8-size and 12-size counterexamples.**

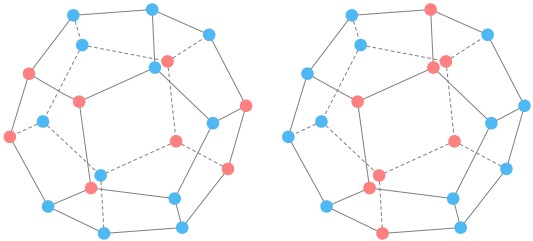

A pair of 8-size geometric graphs (red nodes) and a pair of 12-size geometric graphs (blue nodes) that are counterexamples are shown in the figure above.

For the 8-size counterexample, note that it is actually obtained by carefully inserting two nodes into the 6-size counterexample mentioned just before. We note that there are **2** kinds of nodes in this counterexample, and the numbers of nodes of each kind are 4, 4 respectively.

For the 12-size counterexample, it is actually obtained by carefully deleting two nodes from the 14-size counterexample (corresponds to the way to obtain the 8-size counterexample) mentioned just before. There are **4** kinds of nodes in this counterexample, and the numbers of nodes of each kind are 4, 4, 2, 2 respectively.

**Two pairs of 10-size counterexamples.**

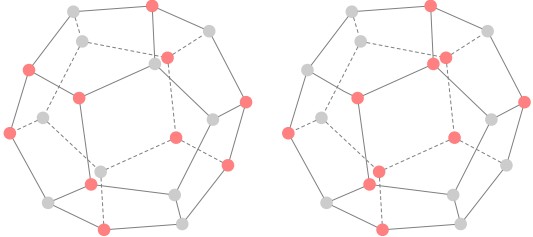

The above figure shows the first pair of 10-size counterexamples. There are **3** kinds of nodes and the numbers of nodes of each kind are 4, 4, 2 respectively.

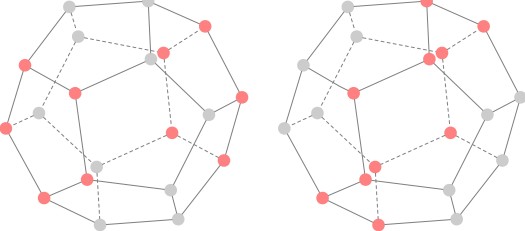

The above figure shows the second pair of 10-size counterexamples. There is actually only **1** kind of nodes, all the nodes of both graphs have the same unordered distance list. It is interesting because

there are two regular pentagons in the left graph and a ring in the right graph, which means that the two graphs are non-congruent, and it is quite similar to the case shown in Figure 1, where there are two equilateral triangles in the left graph and also a ring in the right graph.

In the end, we note that the above counterexamples are probably all the counterexamples one can sample from a single regular polyhedron.

## A.2 Counterexamples That Hold After Some Continuous Transformation

Inspired by the intuition mentioned in Appendix A.1, we can also sample nodes from the combination of regular polyhedrons, which also have great symmetry but with more nodes.

**Cube + regular octahedron.**

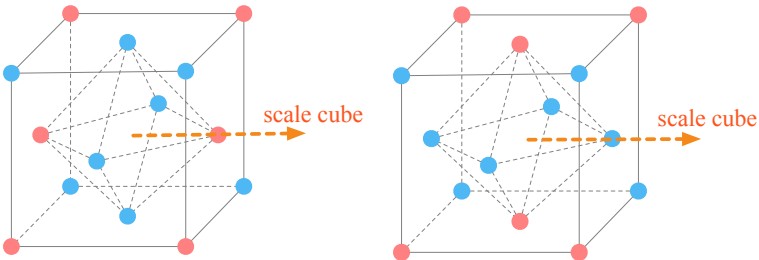

The first two counterexamples are sampled from the combination of a cube and a regular octahedron. We combine them in the following way: Bring the center of both polyhedrons together, and align the vertex of the regular octahedron, the center of the cube's face, and the center of the polyhedrons in a straight line. We do not limit the **relative size** of the two polyhedrons: one can scale the cube to any size as long as the constraints are not broken. In this way, a *small* family of counterexamples is given (we say small here because the dimension of the transformation space is small).

*Proof of the red case.* Let the side length of the cube be $2a$, the diagonal of the regular octahedron be $2b$. Initially, all nodes are labeled $l_0$. We now simulate Vanilla DisGNNs on both graphs.

At the first iteration, there are actually two kinds of unordered distance lists for both of the graphs: For the vertexes on the regular octahedron, the list is $\left\{((2b, l_0), 1), ((\sqrt{3a^2 + b^2 + 2ab}, l_0), 2), \quad ((\sqrt{3a^2 + b^2 - 2ab}, l_0), 2)\right\}$. For the vertexes on the cube, the list is $\left\{((2a, l_0), 1), ((2\sqrt{2}a, l_0), 1), ((2\sqrt{3}a, l_0), 1), ((\sqrt{3a^2 + b^2 + 2ab}, l_0), 1), ((\sqrt{3a^2 + b^2 - 2ab}, l_0), 1)\right\}$. Thus they will be labeled differently at this step, let the labels be $l_1$ and $l_2$ respectively.

At the second iteration, the lists of all the vertexes on the regular octahedron from both graphs are still the same, i.e. $\left\{((2b, l_1), 1), ((\sqrt{3a^2 + b^2 + 2ab}, l_2), 2), ((\sqrt{3a^2 + b^2 - 2ab}, l_2), 2)\right\}$, as well as the lists of all the vertexes on the cube from both graphs, i.e. $\left\{((2a, l_2), 1), ((2\sqrt{2}a, l_2), 1), ((2\sqrt{3}a, l_2), 1), ((\sqrt{3a^2 + b^2 + 2ab}, l_1), 1), ((\sqrt{3a^2 + b^2 - 2ab}, l_1), 1)\right\}$.

Since Vanilla DisGNNs cannot subdivide the vertexes at the second step, they can still not at the following steps. So it cannot distinguish the two geometric graphs. But they are non-congruent: on the left graph, the nodes on the regular octahedron can only form isosceles triangles with the nodes on the face diagonal of the cube. On the right graph, they can only form isosceles triangles with nodes on the cubes that are on the same side. We note that the counterexample in Figure 1 is not a special case of this, because on the left graph of this case all the nodes are on the same surface, but none of graphs in Figure 1 have all nodes on the same surface.

*Proof of the blue case.* The proof of the blue case is quite similar to that of the red case; thus, we omit the proof here. Note that the labels will stabilize after the first iteration. At the end of Vanilla DisGNN, there are **2** kinds of nodes, each kind has 4 nodes. And the two geometric graphs are

non-congruent because the plane formed by four nodes on the regular octahedron is perpendicular to the plane formed by four nodes on the cube in the left graph, and is not in the right graph.

**2 cubes.**

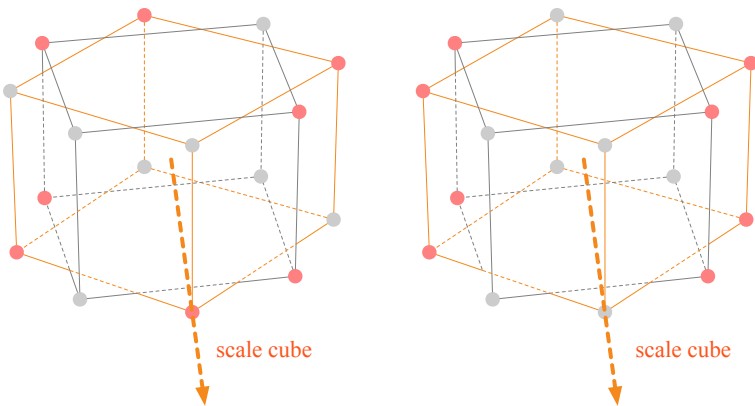

The second case is the combination of 2 cubes. We combine them in the following way: Let the center of the two cubes coincide as well as the axis formed by the centers of two opposite faces. Also, we do not limit the **relative size** of the two cubes: one can scale the cube to any size as long as the constraints are not broken. The proof of this case is quite similar to that in *Cube + regular octahedron*; thus, we omit the proof here. Note that the labels will stabilize after the first iteration. If the sizes of the two cubes are the same, then at the end of Vanilla DisGNN, there is only **1** kind of node. If not, there are **2** kinds of nodes, each kind has 4 nodes.

### A.3 Families of Counterexamples

In this section, we will prove that any one of the counterexamples given in Appendix A.1 can be **augmented** to a family. To facilitate the discussion, we will introduce some notations.

**Notation.** Consider an arbitrary counterexample $(G_{ori}^L, G_{ori}^R)$ given in Appendix A.1. It is worth noting that for each graph, the nodes are sampled from a regular polyhedron and are located on the same spherical surface. We can denote the geometric graph on this sphere as $G_{ori,r}$, where $r$ represents the radius of the sphere. Since the nodes are sampled from some regular polyhedron (denoted by $G_{all,r}$), there must be nodes that aren't sampled, which we call as complementary nodes. We denote the corresponding geometric graph as $G_{com,r}$. Given two graphs $G_{Q_1,r_1}, G_{Q_2,r_2}$ where $Q_i \in \{ori, com, all\}$ for $i \in [2]$, we say that we **align** them if we make sure that the spherical centers of the two geometric graphs coincide and one can get $G_{Q_2,r_2}$ by just scaling all the nodes of $G_{Q_2,r_1}$ along the spherical center. We use $\mathcal{L} = \{(v, l_v) \mid v \in V\}$ to represent a **label state** for some graph $G$, and call it a **stable** state if Vanilla DisGNN cannot further refine the labels of the graph $G$ with initial label $\mathcal{L}$. We say that $\mathcal{L}$ is the **final** state of $G$ if it is the label state obtained by performing Vanilla DisGNN on $G$ with no initial labels for infinite rounds. Note that final state is a special stable state. We denote the final state of $G_{ori}$ and $G_{com}$ as $\mathcal{L}_{ori}$ and $\mathcal{L}_{com}$ respectively.

Now, let us introduce an augmentation method for arbitrary counterexample $(G_{ori}^L, G_{ori}^R)$ shown in Appendix A.1. For a given set $\mathcal{QR}_k = \{(Q_1, r_1), (Q_2, r_2), ..., (Q_k, r_k)\}$ where $r_i > 0, Q_i \in \{ori, com, all\}$ and $r_i \neq r_j$ for $i \neq j$, we get the pair of augmented geometric graphs $(G^L, G^R) = \mathrm{AUG}_{\mathcal{QR}_k}(G_{ori}^L, G_{ori}^R)$ by the following steps (take $G^L$ for example):

1. Generate $k$ geometric graphs $G_{Q_1,r_1}^L, i \in [k]$ according to $\mathcal{QR}_k$.
2. Align the $k$ geometric graphs.
3. Combine the $k$ geometric graphs $G_{Q_i,r_i}^L, i \in [k]$ to obtain $G^L$.

We give an example of the augmentation in Figure 3. Now we give our theorem.

**Theorem A.1.** *Consider an arbitrary counterexample $(G_{ori}^L, G_{ori}^R)$ given in Appendix A.1. Let $\mathcal{QR}_k = \{(Q_1, r_1), (Q_2, r_2), \ldots, (Q_k, r_k)\}$ be an arbitrary set, where $Q_i \in \{ori, com, all\}$ and*

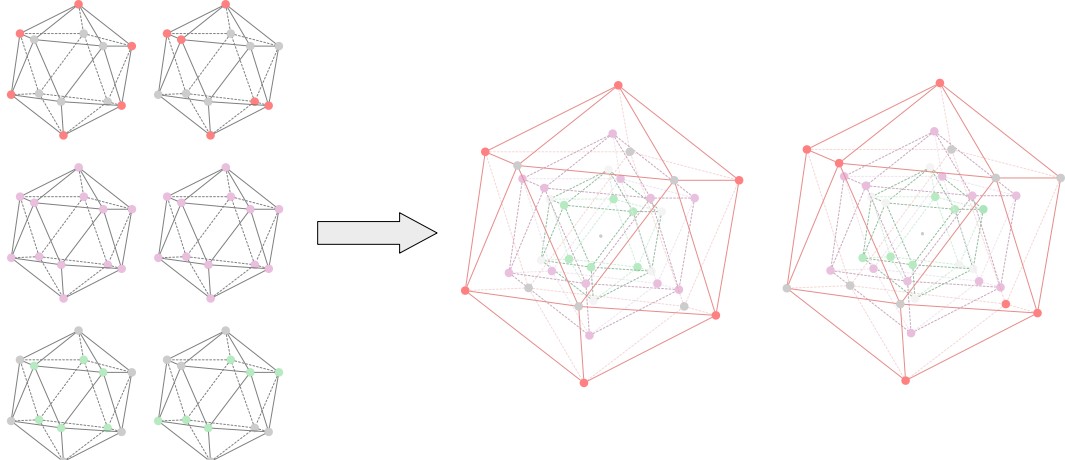

Figure 3: An example of the augmentation. In this example, $(G^L_{ori}, G^R_{ori})$ are from Figure 1, and $\mathcal{QR}_3 = \{(ori, r_r), (all, r_p), (com, r_g)\}$ (with $r, p, g$ representing red, purple and green respectively). Nodes with different colors indicate that they are derived from different $G_{Q,r}$ generations. The augmented graphs $(G^L, G^R) = \text{AUG}_{\mathcal{QR}3}(G^L_{ori}, G^R_{ori})$ consist of all the colored nodes.

$r_i \neq r_j$ for $i \neq j$. If $\exists i \in [k]$ such that $Q_i \neq all$, then the augmented geometric graphs $(G^L, G^R) = \text{AUG}_{\mathcal{QR}_k}(G^L_{ori}, G^R_{ori})$ form a counterexample.

Before presenting the proof of the theorem, we will introduce several lemmas that will be used in the proof.

**Lemma A.2.** *If Vanilla DisGNN cannot distinguish $(G^L, G^R)$ with some initial label $(\mathcal{L}^L, \mathcal{L}^R)$, then Vanilla DisGNN cannot distinguish $(G^L, G^R)$ without initial label (with identical labels).*

*Proof of Lemma A.2.* We use $l^t_u$ to represent the label of $u$ after $t$ iterations of Vanilla DisGNN without initial labels, and use $l^{\mathcal{L},t}_u$ to represent the label of $u$ after $t$ iterations of Vanilla DisGNN with initial label $\mathcal{L}$. We use $l^\infty_u$ and $l^{\mathcal{L},\infty}_u$ to represent the corresponding final label. In order to prove this lemma, we first prove a conclusion: **for arbitrary graphs $G = (V, E)$ and arbitrary initial label $\mathcal{L}$, let $u, v$ be two arbitrary nodes in $V$, then we have $\forall t \in \mathbb{Z}^+, l^t_u \neq l^t_v \rightarrow l^{\mathcal{L},t}_u \neq l^{\mathcal{L},t}_v$.** We'll prove the conclusion by induction.

- $t = 1$ holds.

  If $\left(l^0_u, \{\!\{(d_{us}, l^0_s) \mid s \in N(u)\}\!\}\right) \neq \left(l^0_v, \{\!\{(d_{vs}, l^0_s) \mid s \in N(v)\}\!\}\right)$, then we have $\{\!\{d_{us} \mid s \in N(u)\}\!\} \neq \{\!\{d_{vs} \mid s \in N(v)\}\!\}$, since $l^0_s$ are all identical for $s \in V$. Then it is obvious that $\left(l^{\mathcal{L},0}_u, \{\!\{(d_{us}, l^{\mathcal{L},0}_s) \mid s \in N(u)\}\!\}\right) \neq \left(l^{\mathcal{L},0}_v, \{\!\{(d_{vs}, l^{\mathcal{L},0}_s) \mid s \in N(v)\}\!\}\right)$.

- For $\forall k \in \mathbb{Z}^+$, $t = k$ holds $\Rightarrow t = k + 1$ holds.

  Assume $t = k + 1$ does not hold, i.e., $\exists u, v \in V$, $l^{(k+1)}_u \neq l^{(k+1)}_v$ but $l^{\mathcal{L},(k+1)}_u = l^{\mathcal{L},(k+1)}_v$. We can derive from $l^{\mathcal{L},(k+1)}_u = l^{\mathcal{L},(k+1)}_v$ that

  $$l^{\mathcal{L},(k+1)}_u = l^{\mathcal{L},(k+1)}_v$$
  $$\Rightarrow \left(l^{\mathcal{L},k}_u, \{\!\{(d_{us}, l^{\mathcal{L},k}_s) \mid s \in N(u)\}\!\}\right) = \left(l^{\mathcal{L},k}_v, \{\!\{(d_{vs}, l^{\mathcal{L},k}_s) \mid s \in N(v)\}\!\}\right)$$
  $$\Rightarrow l^{\mathcal{L},k}_u = l^{\mathcal{L},k}_v \text{ and } \{\!\{(d_{us}, l^{\mathcal{L},k}_s) \mid s \in N(u)\}\!\} = \{\!\{(d_{vs}, l^{\mathcal{L},k}_s) \mid s \in N(v)\}\!\}. \quad (13)$$

Since $t = k$ holds, for $\forall s_1, s_2$, we have

$$l^{\mathcal{L},k}_{s_1} = l^{\mathcal{L},k}_{s_2}$$
$$\Rightarrow l^k_{s_1} = l^k_{s_2}.$$

Together with Equation (13), we have $l^k_u = l^k_v$ and $\{\!\{(d_{us}, l^k_s) \mid s \in N(u)\}\!\} = \{\!\{(d_{vs}, l^k_s) \mid s \in N(v)\}\!\}$. This means $l^{(k+1)}_u = l^{(k+1)}_v$, which is contradictory to assumptions.

This means that the conclusion holds for $\forall t \in \mathbb{Z}^+$, as well as $t = \infty$. In other words, specifying an initial label $\mathcal{L}$ will only result in the subdivision of the final labels.

Back to Lemma A.2, if Vanilla DisGNN cannot distinguish $(G^L, G^R)$ even when the final labels are furthermore subdivided, it can still not when the final labels are not subdivided, i.e., cannot distinguish $(G^L, G^R)$ without initial labels. $\qquad\square$

**Lemma A.3.** *Consider an arbitrary counterexample $(G_{ori}^L, G_{ori}^R)$ given in Appendix A.1. We have that $(\mathcal{L}_{ori}^L \cup \mathcal{L}_{com}^L, \mathcal{L}_{ori}^R \cup \mathcal{L}_{com}^R)$ is a pair of **stable states** of $(G_{all}^L, G_{all}^R)$. Denote the labels as $\mathcal{L}_{all}^L$ and $\mathcal{L}_{all}^R$ respectively. Moreover, we have that Vanilla DisGNN cannot distinguish $(G_{all}^L, G_{all}^R)$ with initial labels $\mathcal{L}_{all}^L$ and $\mathcal{L}_{all}^R$.*

*Proof of Lemma A.3.* Since the number of situations is finite, this lemma can be directly verified using a computer program. It is worth noting that we have also included the **verification program** in our code. $\qquad\square$

This lemma allows us to draw several useful prior conclusions. In the graph $G_{all}$, the node set $V_{all}$ can be split into two subsets by definition, namely $V_{ori}$ and $V_{com}$. Now for arbitrary node $u$ from $G_{all}^L$ with initial label $\mathcal{L}_{all}^L$ and $v$ from $G_{all}^R$ with initial label $\mathcal{L}_{all}^R$, if the initial labels of node $u$ and node $v$ are the same, they must be in the same subset, i.e., respectively in $V_{ori}^L$ and $V_{ori}^R$ or $V_{com}^L$ and $V_{com}^R$. Without loss of generality, let us assume that both $u$ and $v$ are in $V_{ori}$. Since $(\mathcal{L}_{all}^L, \mathcal{L}_{all}^R)$ is a stable state of $(G_{all}^L, G_{all}^R)$, we can obtain the following equation:

$$\left(l_u, \{\!\{(d_{us}, l_s) \mid s \in V_{all}^L\}\!\}\right) = \left(l_v, \{\!\{(d_{vs}, l_s) \mid s \in V_{all}^R\}\!\}\right). \tag{14}$$

Since $(\mathcal{L}_{ori}^L, \mathcal{L}_{ori}^R)$ is a stable state of the induced subgraphs $(G_{ori}^L, G_{ori}^R)$ where the node sets are $(V_{ori}^L, V_{ori}^R)$, we can obtain the following equation:

$$\left(l_u, \{\!\{(d_{us}, l_s) \mid s \in V_{ori}^L\}\!\}\right) = \left(l_v, \{\!\{(d_{vs}, l_s) \mid s \in V_{ori}^R\}\!\}\right). \tag{15}$$

Notice that in Equation (14, 15), we can obtain a new equation by replacing $V_{ori}^L$ or $V_{all}^L$ with $(V_{ori}^L - u)$ or $(V_{all}^L - u)$ and doing the same for the $v$ side, while keeping the equation valid, since $l_u = l_v$. By subtracting Equation (15) from Equation (14), we obtain the following equation:

$$\left(l_u, \{\!\{(d_{us}, l_s) \mid s \in V_{com}^L\}\!\}\right) = \left(l_v, \{\!\{(d_{vs}, l_s) \mid s \in V_{com}^R\}\!\}\right). \tag{16}$$

It is important to note that the same conclusion holds if both $u$ and $v$ are from type *com*. These equations provide valuable **prior knowledge** that we can use to prove Theorem A.1.

Now, let us prove Theorem A.1. Our proof is divided into four steps:

1. Construct an initial state $(\mathcal{L}^L, \mathcal{L}^R)$ for the augmented geometric graphs $(G^L, G^R)$.
2. Prove that $(\mathcal{L}^L, \mathcal{L}^R)$ is a pair of stable states for $(G^L, G^R)$.
3. Explain that Vanilla DisGNN cannot distinguish $(G^L, G^R)$ with initial labels $(\mathcal{L}^L, \mathcal{L}^R)$.
4. Explain that $(G^L, G^R)$ are non-congruent.

By applying Lemma A.2, we can get the conclusion from step 2 and 3 that Vanilla DisGNN cannot distinguish $(G^L, G^R)$ without initial labels. Since $(G^L, G^R)$ are non-congruent (as established in step 4), it follows that $(G^L, G^R)$ is a counterexample.

*Proof of Theorem A.1.*

**Step 1.** We first construct an initial state for the augmented graphs $(G^L, G^R)$.

For simplicity, we omit the superscripts, as the rules are the same for both graphs. For each layer (i.e., the sphere of some radius) of the graph $G$, we label the nodes with $\mathcal{L}_{tp}$ if the layer is of type $tp$, for $tp \in \{ori, com, all\}$. Note that we also ensure that the labels in different layers are distinct.

**Step 2.** Now we prove that $(\mathcal{L}^L, \mathcal{L}^R)$ is a stable state for $(G^L, G^R)$. This is equivalent to proving that for arbitrary node $u$ in $(G^L, \mathcal{L}^L)$ and node $v$ in $(G^R, \mathcal{L}^R)$, if $l_u^0 = l_v^0$, then $l_u^1 = l_v^1$.

Since $G^L$ is considered as a complete distance graph by DisGNNs, the neighbors of node $u$ are all the nodes in $G^L$ except itself. We denote the neighbor nodes from the layer with radius $r_i$ by $N_{r_i}(u)$.

This is similar for node $v$. Now we need to prove that $\left(l_u^0, \{\!\{(d_{us}, l_s^0) \mid s \in \bigcup_{i \in [k]} N_{r_i}(u)\}\!\}\right) = \left(l_v^0, \{\!\{(d_{vs}, l_s^0) \mid s \in \bigcup_{i \in [k]} N_{r_i}(v)\}\!\}\right)$. Since $l_u^0 = l_v^0$, we only need to prove that $\{\!\{(d_{us}, l_s^0) \mid s \in \bigcup_{i \in [k]} N_{r_i}(u)\}\!\} = \{\!\{(d_{vs}, l_s^0) \mid s \in \bigcup_{i \in [k]} N_{r_i}(v)\}\!\}$.

We split the multiset $\{\!\{(d_{us}, l_s^0) \mid s \in \bigcup_{i \in [k]} N_{r_i}(u)\}\!\}$ into $k$ multisets, namely $\{\!\{(d_{us}, l_s^0) \mid s \in N_{r_i}(u)\}\!\}, i \in [k]$, and do the same for the multiset of node $v$. Our goal is to prove that $\{\!\{(d_{us}, l_s^0) \mid s \in N_{r_i}(u)\}\!\} = \{\!\{(d_{vs}, l_s^0) \mid s \in N_{r_i}(v)\}\!\}$ for each $i \in [k]$.

Let the coordinates of node $s$ in a spherical coordinate system be $(r_s, \theta_s, \phi_s)$. Since nodes $u$ and $v$ have the same initial label, they must be from the same layer, meaning that $r_u = r_v$. Additionally, we always realign the coordinate systems of $G^L$ and $G^R$ to ensure that the direction of the polyhedra is the same. The distance between node $u$ and node $s$ in a spherical coordinate system is given by $\sqrt{r_u^2 + r_s^2 - 2r_u r_s \big( \sin\theta_u \sin\theta_s \cos(\phi_u - \phi_s) + \cos\theta_u \cos\theta_s \big)}$. We denote the angle term $\big( \sin\theta_u \sin\theta_s \cos(\phi_u - \phi_s) + \cos\theta_u \cos\theta_s \big)$ by $\Theta_{us}$ for simplicty.

For each value of $r_i$, it can be one of three types: *ori*, *com*, or *all*. Similarly, nodes $u$ and $v$ can belong to one of three categories: *ori*, *com*, *all*. Regardless of the combination, these situations can always be found in the Equations (14, 15, 16) derived from Lemma A.3 and can produce the following conclusion:

$$\left(l_u^0, \{\!\{(\Theta_{us}, l_s^0) \mid s \in N_{r_i}(u)\}\!\}\right) = \left(l_v^0, \{\!\{(\Theta_{vs}, l_s^0) \mid s \in N_{r_i}(v)\}\!\}\right).$$

Then we have:

$$\left(l_u^0, \{\!\{(\sqrt{r_u^2 + r_i^2 - 2r_u r_i \Theta_{us}}, l_s^0) \mid s \in N_{r_i}(u)\}\!\}\right)$$
$$= \left(l_v^0, \{\!\{(\sqrt{r_v^2 + r_i^2 - 2r_v r_i \Theta_{vs}}, l_s^0) \mid s \in N_{r_i}(v)\}\!\}\right)$$

since $r_u = r_i$. This means that for all values of $r_i$, $\{\!\{(d_{us}, l_s^0) \mid s \in N_{r_i}(u)\}\!\} = \{\!\{(d_{vs}, l_s^0) \mid s \in N_{r_i}(v)\}\!\}$. By merging all the multisets of radius $r_i$, we can get $\{\!\{(d_{us}, l_s^0) \mid s \in \bigcup_{i \in [k]} N_{r_i}(u)\}\!\} = \{\!\{(d_{vs}, l_s^0) \mid s \in \bigcup_{i \in [k]} N_{r_i}(v)\}\!\}$, which concludes the proof.

**Step 3.** Since the stable states $(\mathcal{L}^L, \mathcal{L}^R)$ are obtained by assigning the stable states of each layer, i.e. $\mathcal{L}_{com}$, $\mathcal{L}_{ori}$ or $\mathcal{L}_{all}$, that cannot be distinguished by Vanilla DisGNN, the histogram of both graphs are exactly the same, which means that Vanilla DisGNN cannot distinguish the two graphs.

**Step 4.** Notice that the construction of an isomorphic mapping requires that the radius of the layer of node $u$ in $G^L$ and node $v$ in $G^R$ must be the same. If there exists a pair of layers in the two geometric graphs that are non-congruent, then the two geometric graphs are non-congruent. According to the definition of $(G^L, G^R)$, such layers always exist, therefore the two geometric graphs are non-congruent. $\qquad\square$

# B Proofs

## B.1 Proof of Proposition 6.3

The main proof of our proposition is to construct the key procedures in GemNet [Gasteiger et al., 2021] using basic message passing layers of 3-E/F-DisGNN, and the whole model can be constructed by stacking these message passing layers up. Since the $k$-E-DisGNN and $k$-F-DisGNN share the same initialization and output blocks and have similar update procedure (they can both learn 4-order geometric information), we mainly focus on the proof of E version, and one can check easily that the derivation for F version also holds.

**Basic methods.** Assume that we want to learn $\mathcal{O}_{\text{tar}} = f_{\text{tar}}(\mathcal{I}_{\text{tar}})$ with our function $\mathcal{O}_{\text{fit}} = f_{\text{fit}}(\mathcal{I}_{\text{fit}})$. In our proof, the form of $\mathcal{O}_{\text{tar}}$ and $\mathcal{O}_{\text{fit}}$, as well as the form of $\mathcal{I}_{\text{tar}}$ and $\mathcal{I}_{\text{fit}}$, are quite different. For example, consider the case where $\mathcal{O}_{\text{fit}}$ is an embedding $h_{abc}$ for a 3-tuple $abc$ and $\mathcal{I}_{\text{fit}} = \mathcal{I}_{\text{fit}}^{(abc)}$ contains the information of all the neighbors of $abc$, while $\mathcal{O}_{\text{tar}}$ is an embedding $m_{ab}$ for a 2-tuple $ab$ and $\mathcal{I}_{\text{tar}} = \mathcal{I}_{\text{tar}}^{(ab)}$ contains the information of all the neighbors of $ab$. Therefore, directly learning functions by $f_{\text{fit}}$ that produces exactly the same output as $f_{\text{tar}}$ is inappropriate. Instead, we will learn functions that can calculate several different outputs in a way $f_{\text{tar}}$ does and appropriately embeds them into the output of $f_{\text{fit}}$. For example, we still consider the case mentioned before, and we want to learn a function $f_{\text{fit}}$ that can extract $\mathcal{I}_{\text{tar}}^{(ab)}, \mathcal{I}_{\text{tar}}^{(ac)}, \mathcal{I}_{\text{tar}}^{(bc)}, \mathcal{I}_{\text{tar}}^{(ba)}, \mathcal{I}_{\text{tar}}^{(ca)}, \mathcal{I}_{\text{tar}}^{(cb)}$ from $\mathcal{I}_{\text{fit}}^{(abc)}$ respectively, and calculates $m_{ab}, m_{ac}, m_{bc}, m_{ba}, m_{ca}, m_{cb}$ with these information like $f_{\text{tar}}$, and embed them into the output $h_{abc}$ in an injective way.

Since we realize $f_{\text{fit}}$ as a universal function approximator (such as MLPs and deep multisets), $f_{\text{fit}}$ can always learn the function we want. So the only concern is that **whether we can extract exact $\mathcal{I}_{\text{tar}}$ from $\mathcal{I}_{\text{fit}}$**, i.e., whether there exists an injective function $(\mathcal{I}_{\text{tar}}^{(1)}, \mathcal{I}_{\text{tar}}^{(2)}, ..., \mathcal{I}_{\text{tar}}^{(n)}) = f_{\text{ext}}(\mathcal{I}_{\text{fit}})$. We will mainly discuss about this in our proof.

**Notations.** We use the superscript $G$ to represent functions in GemNet and $3E$ to represent functions in 3-E-DisGNN, and use $\mathcal{I}$ with the same superscript and subscript to represent the input of a function. We use the superscript $(a_1 a_2 .. a_k)$ to represent some input $\mathcal{I}$ if it's for tuple $(a_1 a_2 .. a_k)$. If there exists an injective function $\mathcal{I}_{\text{tar}} = f_{\text{ext}}(\mathcal{I}_{\text{fit}})$, then we denote it by $\mathcal{I}_{\text{fit}} \to \mathcal{I}_{\text{tar}}$, meaning that $\mathcal{I}_{\text{tar}}$ can be derived from $\mathcal{I}_{\text{fit}}$. If some geometric information $\mathcal{I}_{\text{geo}}$ (such as distance and angles) is contained in the distance matrix of a tuple $a_1 a_2 .. a_k$, then we denote it by $\mathcal{I}_{\text{geo}} \in a_1 a_2 .. a_k$. For simplicity, We omit all the time superscript $t$ if the context is clear.

### B.1.1 Construction of Embedding Block

**Initialization of directional embeddings.** GemNet initialize all the two-tuples $m$ (also called directional embeddings) at the embedding block by the following paradigm

$$m_{ab} = f_{\text{init}}^{\text{G}}(z_a, z_b, d_{ab}). \tag{17}$$

At the initial step, 3-E-DisGNN follows the paradigm outlined below:

$$h_{abc} = f_{\text{init}}^{\text{3E}}(z_a, z_b, z_c, d_{ab}, d_{ac}, d_{bc}). \tag{18}$$

Then we have

$$\mathcal{I}_{\text{init}}^{\text{3E},(abc)} \to (\mathcal{I}_{\text{init}}^{\text{G},(ab)}, \mathcal{I}_{\text{init}}^{\text{G},(ac)}, \mathcal{I}_{\text{init}}^{\text{G},(bc)}, \mathcal{I}_{\text{init}}^{\text{G},(ba)}, \mathcal{I}_{\text{init}}^{\text{G},(ca)}, \mathcal{I}_{\text{init}}^{\text{G},(cb)}),$$

meaning that we can extract all the information to calculate $m_{ab}, m_{ac}, m_{bc}, m_{ba}, m_{ca}, m_{cb}$ from the input of $f_{\text{init}}^{\text{3E}}$. And thanks to the universal property of $f_{\text{init}}^{\text{3E}}$, we can approximate a function that accurately calculates these variables and injectively embeds them into $h_{abc}$, such that $h_{abc} \to (m_{ab}, m_{ac}, m_{bc}, m_{ba}, m_{ca}, m_{cb})$.

**Initialization of atom embeddings.** GemNet initializes all the one-tuples $u_i$ (also called atom embeddings) at the embedding block simply by passing atomic number $z_i$ through an embedding layer.

Note that since we can learn all the $m_{ij}$ and embed them into $h_{abc}$ by $f_{\text{init}}^{\text{3E}}$, it is also possible to learn all the $u_i, i \in \{a, b, c\}$ and embed them into $h_{abc}$ at the same time with some function, since $\mathcal{I}_{\text{init}}^{\text{3E},(abc)}$ contains $z_i, i \in \{a, b, c\}$. Therefore, $h_{abc} \to (u_a, u_b, u_c)$ holds.

**Initialization of geometric information.** It is an important observation that since $f_{\text{init}}^{3E}$ take all the pair-wise distance within 3-tuple $abc$ as input, all the geometric information within the tuple can be included in $h_{abc}$. This makes geometric information rich in the 3-tuple embedding $h_{abc}$.

To remind the readers, now we prove that there exists a function $f_{\text{init}}^{3E}$ which can correctly calculate $m_{ij}$ ($i \neq j, i,j \in \{a,b,c\}$), $u_i$ ($i \in \{a,b,c\}$) and all the geometric information $\mathcal{I}_{\text{geo}}$ within the triangle $abc$, and injectively embed them into $h_{abc}$:

$$h_{abc} \to \Big( \big( m_{ij} \mid i \neq j, i,j \in \{a,b,c\} \big), \big( u_i \mid (i \in \{a,b,c\}) \big), \big( \mathcal{I}_{\text{geo}} \mid \mathcal{I}_{\text{geo}} \in abc \big) \Big).$$

### B.1.2  Construction of Atom Embedding Block

**Atom embedding block.** GemNet updates the 1-tuple embeddings $u$ by summing up all the relevant 2-tuple embeddings $m$ in the atom emb block. The process can be formulated as

$$u_a = f_{\text{atom}}^{G} \big( \{\!\{ (m_{ka}, e_{\text{RBF}}^{(ka)}) \mid k \in [N] \}\!\} \big). \tag{19}$$

Since this function involves the process of pooling, it cannot be learned by $f_{\text{init}}^{3E}$. However, it can be easily learned by the basic message passing layers of 3-E-DisGNN $f_{\text{MP}}^{3E}$, which is formulated as

$$h_{abc} = f_{\text{MP}}^{3E} \big( h_{abc}, \{\!\{ (h_{kbc}, e_{ak}) \mid k \in [N] \}\!\}, \{\!\{ (h_{akc}, e_{bk}) \mid k \in [N] \}\!\}, \{\!\{ (h_{abk}, e_{ck}) \mid k \in [N] \}\!\} \big), \tag{20}$$

$$\text{where } e_{ab} = f_{\text{e}}^{3E} \big( d_{ab}, \{\!\{ h_{kab} \mid k \in [N] \}\!\}, \{\!\{ h_{akb} \mid k \in [N] \}\!\}, \{\!\{ h_{abk} \mid k \in [N] \}\!\} \big). \tag{21}$$

We now want to learn a function $f_{\text{MP}}^{3E}$ that updates the $u_a, u_b, u_c$ embedded in $h_{abc}$ like $f_{\text{atom}}^{G}$ and keep the other variables and information unchanged. Note that $h_{abc}$ is the first input of $f_{\text{MP}}^{3E}$, so all the old information is maintained. As what we talked about earlier, the main focus is to check whether the information to update $u$ is contained in $f_{\text{MP}}^{3E}$'s input. In fact, the following derivation holds

$$\mathcal{I}_{\text{MP}}^{3E,(abc)} \to \{\!\{ (h_{akc}, e_{bk}) \mid k \in [N] \}\!\} \to \{\!\{ h_{akc} \mid k \in [N] \}\!\}$$
$$\to \{\!\{ (m_{ka}, d_{ka}) \mid k \in [N] \}\!\} \to \{\!\{ (m_{ka}, e_{\text{RBF}}^{(ka)}) \mid k \in [N] \}\!\} = \mathcal{I}_{\text{atom}}^{G,(a)}.$$

Note that $d_{ka} \in abc$, and $e_{\text{RBF}}^{(ka)}$ can be calculated from $d_{ka}$. Similarly, we can derive that $\mathcal{I}_{\text{MP}}^{3E,(abc)} \to \mathcal{I}_{\text{atom}}^{G,(b)}$ and $\mathcal{I}_{\text{MP}}^{3E,(abc)} \to \mathcal{I}_{\text{atom}}^{G,(c)}$. This means we can update $u$ in $h$ using a basic message passing layer of 3-E-DisGNN.

### B.1.3  Construction of Interaction Block

**Message passing.** There are two key procedures in GemNet's message passing block, namely two-hop geometric message passing (Q-MP) and one-hop geometric message passing (T-MP), which can be abstracted as follows

$$\text{T} - \text{MP}: \quad m_{ab} = f_{\text{TMP}}^{G} \big( \{\!\{ (m_{kb}, e_{\text{RBF}}^{(kb)}, e_{\text{CBF}}^{(abk)}) \mid k \in [N], k \neq a \}\!\} \big) \tag{22}$$

$$\text{Q} - \text{MP}: \quad m_{ab} = f_{\text{QMP}}^{G} \big( \{\!\{ (m_{k_2 k_1}, e_{\text{RBF}}^{(k_2 k_1)}, e_{\text{CBF}}^{(bk_1 k_2)}, e_{\text{SBF}}^{(abk_1 k_2)}) \mid k_1, k_2 \in [N], k_1 \neq a, k_2 \neq b, a \}\!\} \big) \tag{23}$$

Note that what we need to construct is a function $f_{\text{MP}}^{3E}$ that can update the information about $m_{ij}, i,j \in \{a,b,c\}, i \neq j$ embedded in $h_{abc}$ just like what $f_{\text{TMP}}^{G}$ and $f_{\text{QMP}}^{G}$ do, and keep the other variables and information unchanged. Since it is quite similar among different $m_{ij}$, we will just take the update process of $m_{ab}$ for example.

First, T-MP. For this procedure, the following derivation holds

$$\mathcal{I}_{\text{MP}}^{3E,(abc)} \to \big( h_{abc}, \{\!\{ (h_{kbc}, e_{ak}) \mid k \in [N] \}\!\} \big) \to \{\!\{ (h_{abc}, h_{kbc}, e_{ak}) \mid k \in [N] \}\!\}$$
$$\to \{\!\{ (m_{kb}, d_{kb}, d_{ab}, \phi_{abk}) \mid k \in [N] \}\!\} \to \{\!\{ (m_{kb}, e_{\text{RBF}}^{(kb)}, e_{\text{CBF}}^{(abk)}) \mid k \in [N], k \neq a \}\!\} = \mathcal{I}_{\text{TMP}}^{G,(ab)}.$$

Note that in the derivation above, there is an important conclusion implicitly used: the tuple $(h_{abc}, h_{kbc}, e_{ak})$ actually contains all the geometric information in the 4-tuple $abck$, because the

distance matrix of the four nodes can be obtained from it. Thus $d_{kb}, d_{ab}, \phi_{abk}$ can be obtained from it, and since $e_{\text{RBF}}^{(kb)}$ and $e_{\text{CBF}}^{(abk)}$ are just calculated from these geometric variables, the derivation holds.

And we can exclude the element in the multiset where the index $k = a$, simply because these tuples have different patterns from others.

Second, Q-MP. In Q-MP, the pooling objects consist of two indices (which we call two-order pooling), namely $k_1$ and $k_2$ in Equation (23). One alternative way to do this is two-step pooling, i.e. pool two times and once for one index. For example we can pool index $k_1$ before we pool index $k_2$ as follows:

$$m_{ab} = f_{\text{QMP}'}^{\text{G}}\big(\{\!\{\{\!\{(m_{k_2 k_1}, e_{\text{RBF}}^{(k_2 k_1)}, e_{\text{CBF}}^{(bk_1 k_2)}, e_{\text{SBF}}^{(abk_1 k_2)}) \mid k_1 \in [N], k_1 \neq a\}\!\} \mid k_2 \in [N], k_2 \neq b, a\}\!\}\big). \tag{24}$$

Note that the expressiveness of two-step pooling is not less than two-order pooling, i.e.

$$\{\!\{\{\!\{(m_{k_2 k_1}, e_{\text{RBF}}^{(k_2 k_1)}, e_{\text{CBF}}^{(bk_1 k_2)}, e_{\text{SBF}}^{(abk_1 k_2)}) \mid k_1 \in [N], k_1 \neq a\}\!\} \mid k_2 \in [N], k_2 \neq b, a\}\!\}$$

$$\to \{\!\{(m_{k_2 k_1}, e_{\text{RBF}}^{(k_2 k_1)}, e_{\text{CBF}}^{(bk_1 k_2)}, e_{\text{SBF}}^{(abk_1 k_2)}) \mid k_1, k_2 \in [N], k_1 \neq a, k_2 \neq b, a\}\!\}.$$

Thus, if we use two-step pooling to implement Q-MP, it does not reduce expressiveness. Inspired by this, in order to update $m_{ab}$ in $h_{abc}$ like $f_{\text{QMP}}^{\text{G}}$, we first learn a function by $f_{\text{MP}}^{\text{3E}}$ that calculates an intermediate variable $w_c = f_{\text{inter1}}^{\text{3E}}\big(\{\!\{(m_{ck}, e_{\text{RBF}}^{(ck)}, e_{\text{CBF}}^{(bkc)}, e_{\text{SBF}}^{(abkc)}) \mid k \in [N], k \neq a\}\!\}\big)$ by pooling all the $m_{ck}$ at index $k$, which is feasible because the following derivation holds

$$\begin{aligned}
\mathcal{I}_{\text{MP}}^{\text{3E},(abc)} &\to \big(h_{abc}, \{\!\{(h_{abk}, e_{ck}) \mid k \in [N]\}\!\}\big) \to \{\!\{(h_{abc}, h_{abk}, e_{ck}) \mid k \in [N]\}\!\} \\
&\to \{\!\{(m_{ck}, d_{ck}, d_{bk}, \phi_{abk}, \theta_{abkc}) \mid k \in [N]\}\!\} \\
&\to \{\!\{(m_{ck}, e_{\text{RBF}}^{(ck)}, e_{\text{CBF}}^{(bkc)}, e_{\text{SBF}}^{(abkc)}) \mid k \in [N], k \neq a\}\!\} = \mathcal{I}_{\text{inter}}^{\text{3E},(c)}.
\end{aligned}$$

Note that in the derivation process above, $m_{ck}$ is directly derived from $e_{ck}$ because it contains the information by definition 21. Then we apply another message passing layer $f_{\text{MP}}^{\text{3E}}$ but this time we learn a function that just pools all the $w_c$ in $h_{abc}$ and finally updates $m_{ab}$:

$$\mathcal{I}_{\text{MP}}^{\text{3E},(abc)} \to \{\!\{(h_{abk}, e_{ck}) \mid k \in [N]\}\!\} \to \{\!\{h_{abk} \mid k \in [N]\}\!\} \to \{\!\{w_k \mid k \in [N], k \neq b, a\}\!\}.$$

This means we can realize $f_{\text{QMP}'}^{\text{G}}$ by stacking two message passing layers up.

**Atom self-interaction.** This sub-block actually involves two procedures: First, update atom embeddings $u$ according to the updated directional embeddings $m$. Second, update the directional embeddings $m$ according to the updated atom embeddings $u$. The first step is actually an atom embedding block, which is already realized by our $f_{\text{MP}}^{\text{3E}}$ in Appendix B.1.1. The second procedure can be formulated as

$$m_{ab} = f_{\text{self}-\text{inter}}^{\text{G}}(u_a, u_b, m_{ab}). \tag{25}$$

It is obvious that we can update $m_{ab}$ in $h_{abc}$ by $f_{\text{MP}}^{\text{3E}}$ because the following derivation holds

$$\mathcal{I}_{\text{MP}}^{\text{3E},(abc)} \to h_{abc} \to (u_a, u_b, m_{ab}) = \mathcal{I}_{\text{self}-\text{inter}}^{\text{G},(ab)}.$$

### B.1.4 Construction of Output Block

In GemNet, the final output $t$ is obtained by summing up all the sub-outputs from each interaction block. While it is possible to add additional sub-output blocks to our model, in our proof we only consider the case where the output is obtained solely from the final interaction block.

The following function produces the output of GemNet

$$t = \sum_{a \in [N]} W_{\text{out}}\big(f_{\text{atom}}^{\text{G}}(\{\!\{(m_{ka}, e_{\text{RBF}}^{(ka)}) \mid k \in [N]\}\!\})\big), \tag{26}$$

where $W_{\text{out}}$ is a learnable matrix.

And our output function is

$$t = f_{\text{output}}^{\text{3E}}\big(\{\!\{h_{abc} \mid a, b, c \in [N]\}\!\}\big). \tag{27}$$

We can realize Equation (26) by stacking a message passing layer and an output block. First, the message passing layer updates all the atom embeddings $u$ in $h_{abc}$, and then the output block extracts $u$ from $h$ as follows, and calculates $t$ like GemNet.

$$\mathcal{I}_{\text{output}}^{\text{3E},(abc)} \to \{\!\{h_{aaa} \mid a \in [N]\}\!\} \to \{\!\{u_a \mid a \in [N]\}\!\}.$$

## B.2 Proof of Proposition 6.2

In this section, we will follow the basic method and notations in Appendix B.1. We will mainly prove for 2-F-DisGNN, and since when $k = 2$, $k$-E-DisGNN's update function can implement $k$-F-DisGNN's (See Appendix B.4), the derivations also holds for 2-E-DisGNN. We will use the superscript *D* to represent functions in DimeNet and *2F* to represent functions in 2-F-DisGNN.

### B.2.1 Construction of Embedding Block

DimeNet initializes all the two tuples in the embedding block just like GemNet, which can be formulated as

$$m_{ab} = f_{\text{init}}^{\text{D}}(z_a, z_b, d_{ab}). \tag{28}$$

What 2-F-DisGNN does at the initialization step is

$$h_{ab} = f_{\text{init}}^{\text{2F}}(z_a, z_b, d_{ab}). \tag{29}$$

Now assume we want to learn such a function that can learn both $m_{ab}$ and $m_{ba}$, then embed it into $h_{ab}$. As what we talked about in Appendix B.1, it is possible because the following derivation holds

$$\mathcal{I}_{\text{init}}^{\text{2F},(ab)} \to (\mathcal{I}_{\text{init}}^{\text{D},(ab)}, \mathcal{I}_{\text{init}}^{\text{D},(ba)}).$$

Note that $h_{ab}$ also contains the geometric information, but in this case, just the distance. Different from GemNet, DimeNet does not "track" the 1-tuple embeddings: it does not initialize and update the atom embeddings in the model, and only pool the 2-tuples into 1-tuples in the output block. So there is no need to embed the embeddings of atom $a$ and atom $b$ into $h_{ab}$.

### B.2.2 Construction of Interaction Block

The message passing framework of DimeNet (so-called directional message passing) can be formulated as

$$m_{ab} = f_{\text{MP}}^{\text{D}}\big(\{\!\!\{(m_{ka}, e_{\text{RBF}}^{(ab)}, e_{\text{CBF}}^{(kab)}) \mid k \in [N], k \neq b\}\!\!\}\big). \tag{30}$$

And the message passing framework of 2-F-DisGNN can be formulated as

$$h_{ab} = f_{\text{MP}}^{\text{2F}}\big(h_{ab}, \{\!\!\{(h_{kb}, h_{ak}) \mid k \in [N], k \neq b\}\!\!\}\big). \tag{31}$$

Now we want to learn a function $f_{\text{MP}}^{\text{2F}}$ that can updates the $m_{ab}$ and $m_{ba}$ embedded in $h_{ab}$ like what $f_{\text{MP}}^{\text{D}}$ does. We need to check if $f_{\text{MP}}^{\text{2F}}$ have sufficient information to update $m_{ab}$ and $m_{ba}$ embedded in its output $h_{ab}$. The derivation holds:

$$\mathcal{I}_{\text{MP}}^{\text{2F},(ab)} \to \{\!\!\{(h_{ab}, h_{kb}, h_{ak}) \mid k \in [N]\}\!\!\} \to \{\!\!\{(m_{ka}, d_{ab}, d_{ka}, \phi_{kab}) \mid k \in [N], k \neq b\}\!\!\}$$
$$\to \{\!\!\{(m_{ka}, e_{\text{RBF}}^{(ab)}, e_{\text{CBF}}^{(kab)}) \mid k \in [N], k \neq b\}\!\!\} = \mathcal{I}_{\text{MP}}^{\text{D},(ab)}.$$

Note that we again used the observation that tuple $(h_{ab}, h_{kb}, h_{ak})$ contains all the geometric information of 3-tuple $abk$. Similarly we can get $\mathcal{I}_{\text{MP}}^{\text{2F},(ab)} \to \mathcal{I}_{\text{MP}}^{\text{D},(ba)}$.

### B.2.3 Construction of Output Block

The output block of DimeNet is quite similar to that of GemNet (Appendix B.1.4), which can be formulated as

$$u_a = f_{\text{atom}}^{\text{D}}\big(\{\!\!\{(m_{ka}, e_{\text{RBF}}^{(ka)}) \mid k \in [N]\}\!\!\}\big), \tag{32}$$

$$t = \sum_{a \in [N]} u_a. \tag{33}$$

This can be realized by stacking a message passing layer of 2-F-DisGNN and its output block up.

The message passing layer of 2-F-DisGNN can learn $u_a$ and $u_b$ and embed it into the output $h_{ab}$ because the following derivation holds

$$\mathcal{I}_{\text{MP}}^{\text{2F},(ab)} \to \{\!\!\{(h_{kb}, h_{ak}) \mid k \in [N]\}\!\!\} \to \{\!\!\{h_{ak} \mid k \in [N]\}\!\!\}$$
$$\to \{\!\!\{(m_{ka}, d_{ka}) \mid k \in [N]\}\!\!\} \to \{\!\!\{(m_{ka}, e_{\text{RBF}}^{(ka)}) \mid k \in [N]\}\!\!\} = \mathcal{I}_{\text{atom}}^{\text{D},(a)}.$$

Similarly we can get $\mathcal{I}_{\mathrm{MP}}^{\mathrm{2F},(ab)} \to \mathcal{I}_{\mathrm{atom}}^{\mathrm{D},(b)}$.

And the output block of 2-F-DisGNN is formulated as follows

$$t = f_{\mathrm{output}}^{\mathrm{2F}}\left(\{\!\{ h_{ab} \mid a,b \in [N] \}\!\}\right). \tag{34}$$

Note that the following derivation holds

$$\mathcal{I}_{\mathrm{output}}^{\mathrm{2F},(ab)} \to \{\!\{ h_{aa} \mid a \in [N] \}\!\} \to \{\!\{ u_a \mid a \in [N] \}\!\}.$$

This means that the output block of 2-F-DisGNN can implement the sum operation in Equation (33).

## B.3  Proof of Theorem 6.4

We first restate the theorem formally.

**Theorem B.1.** *Let $\mathcal{M}(\theta) \in \mathrm{MinMods} = \{$1-round 4-DisGNN, 2-round 3-E-DisGNN, 2-round 3-F-DisGNN$\}$ with parameters $\theta$. Denote $h_m = f_{\mathrm{node}}\left(\{\!\{ h_{\boldsymbol{v}}^T \mid \boldsymbol{v} \in V^k, \boldsymbol{v}_0 = m \}\!\}\right) \in \mathbb{R}^{K'}$ as node $m$'s representation produced by $\mathcal{M}$ where $f_{\mathrm{node}}$ is an injective multiset function, and $\mathbf{x}_m^c$ as node $m$'s coordinates w.r.t. the center. Denote $\mathrm{MLPs}(\theta') : \mathbb{R}^{K'} \to \mathbb{R}$ as a multi-layer perceptron with parameters $\theta'$.*

1. *(**Completeness**) Given arbitrary two geometric graphs $\mathbf{X}_1, \mathbf{X}_2 \in \mathbb{R}^{3 \times n}$, then there exists $\theta_0$ such that $\mathcal{M}(\mathbf{X_1}; \theta_0) = \mathcal{M}(\mathbf{X_2}; \theta_0) \iff \mathbf{X_1}$ and $\mathbf{X_1}$ are congruent.*

2. *(**Universality for scalars**) Let $f : \mathbb{R}^{3 \times n} \to \mathbb{R}$ be an arbitrary continuous, $E(3)$ invariant and permutation invariant function over geometric graphs, then for any compact set $M \subset \mathbb{R}^{3 \times n}$ and $\epsilon > 0$, there exists $\theta_0$ such that $\forall \mathbf{X} \in M, |f(\mathbf{X}) - \mathcal{M}(\mathbf{X}; \theta_0)| \leq \epsilon$.*

3. *(**Universality for vectors**) Let $f : \mathbb{R}^{3 \times n} \to \mathbb{R}^3$ be an arbitrary continuous, $O(3)$ equivariant, permutation and translation invariant function over geometric graphs, let $f_{\mathrm{out}}^{\mathrm{equiv}} = \sum_{m=1}^{|V|} \mathrm{MLPs}(h_m) \mathbf{x}_m^c$, then for any compact set $M \subset \mathbb{R}^{3 \times n}$ and $\epsilon > 0$, there exists $\theta_0$ for $\mathcal{M}$ and $\theta_0'$ for $\mathrm{MLPs}$ such that $\forall \mathbf{X} \in M, |f(\mathbf{X}) - f_{\mathrm{out}}^{\mathrm{equiv}}(\mathbf{X}; \theta_0, \theta_0')| \leq \epsilon$.*

In the forthcoming proof for completeness part, we examine the optimal scenario for $k$-(E/F-)DisGNNs, namely the geometric version $k$-(E/F)WL, where the hyper-parameters such as hidden dimensions are set and parameters $\theta$ of $\mathcal{M}(\theta)$ are learned to ensure that all the relevant functions are hash functions. To simplify the analysis for $k$-EWL, we substitute the edge representation in $k$-E-DisGNN with the edge weight (distance) alone, which is already enough.

**Basic idea for proof of completeness.** We prove the completeness part of the theorem, which demonstrates that the optimal DisGNNs have the ability to distinguish all geometric graphs, through a *reconstruction* method. Specifically, if we can obtain all points' coordinates up to permutation and E(3) translation from the output of DisGNNs, it implies that DisGNNs can differentiate between all geometric graphs.

### B.3.1  Proof of Completeness for One-Round 4-WL

*Proof.* We begin by assuming the existence of four affinely independent nodes, meaning that there exists a 4-tuple of nodes that can form a tetrahedron. If this condition is not met, and all the nodes lie on the same plane, the problem is simplified, as the cloud degenerates into 2D space. We will address this case later in the discussion. We assume the point number is $N$.

Our first step is to prove that we can recover the 4-tuple formed by the four affinely independent nodes from the output of one-round 4-WL, denoted by $\{\!\{ c_{\boldsymbol{i}}^1 \mid \boldsymbol{i} \in [N]^4 \}\!\}$. Since all the tuple colors are initialized injectively according to the tuple's distance matrices, there exists a function $f^0$ such that $f^0(c_{\boldsymbol{i}}^0) = D_{\boldsymbol{i}}$, where $D_{\boldsymbol{i}}$ is the distance matrix of tuple $\boldsymbol{i}$. Note that since the update procedures of the colors are HASH functions, $c_{\boldsymbol{i}}^t = \mathrm{HASH}(c_{\boldsymbol{i}}^{t-1}, C_{0,\boldsymbol{i}}^{t-1}, C_{1,\boldsymbol{i}}^{t-1}, C_{2,\boldsymbol{i}}^{t-1}, C_{3,\boldsymbol{i}}^{t-1})$ where $C_{m,\boldsymbol{i}}^{t-1}$ denotes the color multiset of tuple $\boldsymbol{i}$'s $m$-th neighbors, there also exists a series functions $f^t$ such that $f^t(c_{\boldsymbol{i}}^t) = c_{\boldsymbol{i}}^{t-1}$. This simply means that the latter colors of a tuple will contain all the information of its former colors. Given the fact, we can use the function $f^1 \circ f^0$ to reconstruct the distance matrices of all the tuples, and find the one that represents a tetrahedron geometry (Theorem 3.1). We mark the found tuple with $\boldsymbol{k}$.

Now we prove that one can reconstruct the whole geometry from just $c_{\boldsymbol{k}}^1$.

First, we can reconstruct the 4 points' 3D coordinates (given any arbitrary center and orientation) from $\boldsymbol{k}$'s distance matrix (Theorem 3.1). Formally, there exists a function $f^D(D_{\boldsymbol{k}}) = X_{\boldsymbol{k}}$, where $X_{\boldsymbol{k}} \in \mathbb{R}^{4*3}$ represents the coordinates. And since $f^1 \circ f^0(c_{\boldsymbol{k}}^1) = D_{\boldsymbol{k}}$, we have that $f^1 \circ f^0 \circ f^D(c_{\boldsymbol{k}}^1) = X_{\boldsymbol{k}}$.

The update function of $c_{\boldsymbol{k}}$ is $c_{\boldsymbol{k}}^1 = \mathrm{HASH}(c_{\boldsymbol{k}}^0, C_{0,\boldsymbol{k}}^0, C_{1,\boldsymbol{k}}^0, C_{2,\boldsymbol{k}}^0, C_{3,\boldsymbol{k}}^0)$, where $C_{m,\boldsymbol{k}}^0 = \{\!\{c_{\Phi_m(\boldsymbol{k},j)}^0 \mid j \in [N]\}\!\}, m \in [4]$ and $\Phi_m(\boldsymbol{k}, j)$ replaces the $m$-th element in tuple $\boldsymbol{k}$ with $j$. Since the function is HASH function, we can also reconstruct each of $C_{m,\boldsymbol{k}}^0$ from $c_{\boldsymbol{k}}^1$. In $C_{m,\boldsymbol{k}}^0$, there exists $N$ 4-tuples' initial color, from which we can reconstruct the 4-tuple's distance matrix. Note that given a color $c_{\Phi_m(\boldsymbol{k},j)}^0$ in the multiset, we can reconstruct 3 distances, namely $d(x_j, x_{\boldsymbol{k}_{(m+1)\%4}}), d(x_j, x_{\boldsymbol{k}_{(m+2)\%4}}), d(x_j, x_{\boldsymbol{k}_{(m+3)\%4}})$, where $x_j$ representes the 3D coordinates of point $j$ and $d(x,y)$ calculates the $l_2$ norm of $x, y$.

There is a strong geometric constrain in 3D space: Given the distances between a point and 3 affinely independent points (whose postions are known), one can calculate at most 2 possible positions of the point, and the 2 possible postions are mirror-symmetric relative to the plane formed by the 3 affinely independent points. Moreover, the point is on the plane formed by the 3 affinely independent points iff there is only one solution of the distance equations.

Now since we have already reconstructed $X_{\boldsymbol{k}}$, and $\boldsymbol{k}_{(m+1)\%4}, \boldsymbol{k}_{(m+2)\%4}, \boldsymbol{k}_{(m+3)\%4}$ are affinely independent, we can calculate 2 possible positions of point $j$. And for the whole multiset $C_{m,\boldsymbol{k}}^0$, we can calculate a $N$-size multiset where each element is a pair of possible positions of some point, denoted as $P_{m,\boldsymbol{k}} = \{\!\{\{x_j^{(1)}, x_j^{(2)}\} \mid j \in [N]\}\!\}$. Note that now the possible geometry of the $N$ points form a 0-dimension manifold, with no more than $2^N$ elements.

There are two important properties of $P_{m,\boldsymbol{k}}$:

1. Given arbitrary two elements (pairs) of $P_{m,\boldsymbol{k}}$, denoted as $p_{j_1}$ and $p_{j_2}$, we have $p_{j_1} = p_{j_2}$ or $p_{j_1} \cap p_{j_2} = \emptyset$. These correspond to two possible situations: $j_1$ and $j_2$ are either mirror-symmetric relative to the plane formed by $\boldsymbol{k}_{(m+1)\%4}, \boldsymbol{k}_{(m+2)\%4}, \boldsymbol{k}_{(m+3)\%4}$, or not.

2. Arbitrary element $p_j$ of $P_{m,\boldsymbol{k}}$ has multiplicity value at most 2. This corresponds two possible situations: $j$ either has a mirror-symmetric point relative to the plane formed by $\boldsymbol{k}_{(m+1)\%4}, \boldsymbol{k}_{(m+2)\%4}, \boldsymbol{k}_{(m+3)\%4}$, or does not.

The follows proves that the four 0-dimension manifolds determined by $P_{m,\boldsymbol{k}}, m \in [4]$ can intersect at a unique solution, which is the real geometry.

We first "refine" the multiset $P_{m,\boldsymbol{k}}$. There are 3 kinds of pairs in $P_{m,\boldsymbol{k}}$:

1. The pair $\{x_j^{(1)}, x_j^{(2)}\}$ that have multiplicity value 2. This means that there are two mirror-symmetric points relative to the plane formed by $\boldsymbol{k}_{(m+1)\%4}, \boldsymbol{k}_{(m+2)\%4}, \boldsymbol{k}_{(m+3)\%4}$. So we can ensure that the two points with coordinates $x_j^{(1)}, x_j^{(2)}$ both exist.

2. The pair where $x_j^{(1)} = x_j^{(2)} = x_j$. This means that by solving the 3 distance equations, there is only one solution. So we can uniquely determine the coordinates of point $j$ (and the point is on the plane).

3. The other pairs.

We can determine the real coordinates of the points from the first two kinds of pairs, so we record the real coordinates and delete them from the multiset $P_{m,\boldsymbol{k}}$ (for the first kind, delete both two pairs). We denote the real coordinates determined by $P_{m,\boldsymbol{k}}$ as $A_m$. Now we have 4 preliminarily refined sets, denoted as $P'_{m,\boldsymbol{k}}$. Note that now in each $P'_{m,\boldsymbol{k}}$:

1. Arbitrary two elements intersect at $\emptyset$.

2. Arbitrary element has two distinct possible coordinates, one is real and the other is fake.

3. The two properties above also mean that elements in multiset $\bigcup_{p \in P'_{m,k}} p$ has only multiplicity 1.

Since the real coordinates $A_m$ determined by different $P_{m,k}$ may be not the same, we further refine the $P'_{m,k}$ by removing the pairs that contain coordinates in $(A_0 \cup A_1 \cup A_2 \cup A_3) - A_m$. Note that according to the property of $P'_{m,k}$ mentioned above, each coordinates in $(A_0 \cup A_1 \cup A_2 \cup A_3) - A_m$ will and will only corresponds to one pair in $P'_{m,k}$. Denote the further refined sets as $P''_{m,k}$. After the refinement, each $P''_{m,k}$ has equally $M = N - |A_0 \cup A_1 \cup A_2 \cup A_3|$ elements (pairs), corresponding to $M$ undetermined real coordinates.

We use $p \cap P$ to denote the multiset $\{p' \mid p' \in P, p' \cap p \neq \emptyset\}$. Now we prove that $\exists m \in [4], \exists p \in P''_{m,k}, \exists m' \in [4] - m$, s.t. $|p \cap P''_{m',k}| = 1$ by contradiction.

Note that $\forall m \in [4], \forall p \in P''_{m,k}, \forall m' \in [4] - m, |p \cap P''_{m',k}| \in \{1, 2\}$. That's because each element in $P''_{m,k}$ contains one real coordinates and one fake coordinates, each real/fake coordinates can at most intersect with one pair in $P''_{m',k}$, and there must exist one element in $P''_{m',k}$ that contains the same real coordinates.

So opposite to the conclusion is: $\forall m \in [4], \forall p \in P''_{m,k}, \forall m' \in [4] - m, |p \cap P''_{m',k}| = 2$, we assume this holds. Now there are also two properties:

1. For all $m \in [4]$, elements in multiset $\bigcup_{p \in P''_{m,k}} p$ has multiplicity 1.

2. For all $m \in [4]$, $\bigcup_{p \in P''_{m,k}} p$ with different $m$ shares the same $M$ real coordinates.

So we have that: $\bigcup_{p \in P''_{m,k}} p$ is the same across all $m \in [4]$. Note that for each $p \in P''_{m,k}$, since the two coordinates in $p$ are mirror-symmetric relative to plane $k_{(m+1)\%4}, k_{(m+2)\%4}, k_{(m+3)\%4}$, the midpoint of the two points is on the plane. This means that by averaging all the coordinates in $\bigcup_{p \in P''_{m,k}} p$, we have a center point on the plane $k_{(m+1)\%4}, k_{(m+2)\%4}, k_{(m+3)\%4}$. However, since $\bigcup_{p \in P''_{m,k}} p$ for all $m \in [4]$ are the same, we have one same center point on each four planes $k_{(m+1)\%4}, k_{(m+2)\%4}, k_{(m+3)\%4}$, $m \in [4]$. This contradicts to the assumption that $k_1, k_2, k_3, k_4$ are affinely independent points. So the conclusion that $\exists m \in [4], \exists p \in P''_{m,k}, \exists m' \in [4] - m$, s.t. $|p \cap P''_{m',k}| = 1$ holds.

Since the conclusion holds, we can obtain the $p$ and the unique element $p' \in p \cap P''_{m',k}$. Moreover, we have that $|p' \cap p| = 1$. This is because, the two pairs share one real coordinates, and the fake coordinates must be different (The two planes formed by $k_{(m+1)\%4}, k_{(m+2)\%4}, k_{(m+3)\%4}$ and $k_{(m'+1)\%4}, k_{(m'+2)\%4}, k_{(m'+3)\%4}$ are not the same). So we obtain the real coordinates by $p' \cap p$.

Now we further refine the 4 $P''_{m,k}$ by deleting the pair that contains the real coordinates we just found, which makes each $P''_{m,k}$ have only $M - 1$ pairs now. Then we loop the procedure above to find all the remaining real coordinates.

Note that all the information, including the final geometry, we obtained in the above proof is derived from only $c^1_k$. This means that when generating $c^1_k$, 4-WL injectively embeds the whole geometry using a HASH function. For two non-congruent geometric graph, their 4-tuple's (where 4 nodes are affinely independent) colors will always be different, giving the 4-WL the power to distinguish them.

For the degenerated situation (there does not exist 4 affinely independent nodes), the points form a 2D plane. Then with similar proof idea, we can prove that 3-WL can be universal on 2D plane. Since 3-WL is not stronger than 4-WL, 4-WL can also distinguish all the degenerated cases. □

### B.3.2   Proof of Completeness for Two-Round 3-FWL

*Proof.* As previous, our discussion is under the fact that the point cloud does not degenerates to a 2D or 1D point cloud. Otherwise, the problem is quite trivial.

We first find the tuple containing three nodes that are affinely independent from the 2-round 3-FWL's output $\{c^2_i \mid i \in [N]^3\}$. Denote the founded tuple as $k$. We first calculate the 3 nodes' coordinates according to the $c^0_k$ derived from $c^2_k$.

As the update procedure, $c_{\boldsymbol{k}}^2 = \text{HASH}(c_{\boldsymbol{k}}^1, \{\!\!\{ (c_{\Phi_0(\boldsymbol{k},j)}^1, c_{\Phi_1(\boldsymbol{k},j)}^1, c_{\Phi_2(\boldsymbol{k},j)}^1) \mid j \in N \}\!\!\})$, each tuple $(c_{\Phi_0(\boldsymbol{k},j)}^1, c_{\Phi_1(\boldsymbol{k},j)}^1, c_{\Phi_2(\boldsymbol{k},j)}^1)$ in the multiset contains the distance $d_{\boldsymbol{k}_0,j}, d_{\boldsymbol{k}_1,j}, d_{\boldsymbol{k}_2,j}$. Thus, we can calculate at most two possible coordinates of node $j$, and the two coordinates are mirror-symmetric relative to plane $\boldsymbol{k}$. Like in the proof of 4-WL, we denote the final calculaoed possible coordinates-pair multiset as $P_{\boldsymbol{k}} = \{\!\!\{ \{x_j^{(1)}, x_j^{(2)}\} \mid j \in [N] \}\!\!\}$.

We then further find a node $j_0$ that is not on plane $\boldsymbol{k}$ from the multiset term in $c_{\boldsymbol{k}}^2$, and its colors' tuple is $(c_{\Phi_0(\boldsymbol{k},j_0)}^1, c_{\Phi_1(\boldsymbol{k},j_0)}^1, c_{\Phi_2(\boldsymbol{k},j_0)}^1)$. Note that the three colors are at time-step 1, which means that they already aggregated neighbors' information for one-round. So by repeating the above procedure, we can again get a possible coordinates-pair multiset $P_{\Phi_m(\boldsymbol{k},j_0)}$ from each of $c_{\Phi_m(\boldsymbol{k},j_0)}^1$ where $m \in [3]$.

Note that the four plane, $\boldsymbol{k}, \Phi_0(\boldsymbol{k},j_0), \Phi_1(\boldsymbol{k},j_0), \Phi_2(\boldsymbol{k},j_0)$, do not intersect at a common point. So we can do as what we do in the proof of 4-WL, to repeatedly refine each multiset $P$ and get all the real coordinates, thus reconstruct the whole geometry. $\qquad\square$

### B.3.3 Proof of Completeness for Two-Round 3-E-DisGNN

*Proof.* As previous, our discussion is under the fact that the point cloud does not degenerates to a 2D or 1D point cloud. Otherwise, the problem is quite trivial.

So we first find the tuple containing three nodes that are affinely independent from the 2-round 3-E-DisGNN's output $\{\!\!\{ c_{\boldsymbol{i}}^2 \mid \boldsymbol{i} \in [N]^3 \}\!\!\}$, and refer to it as tuple $\boldsymbol{k}$. Its output color $c_{\boldsymbol{k}}^2$ can derive the three points' coordinates. The update procedure is $c_{\boldsymbol{k}}^2 = \text{HASH}(c_{\boldsymbol{k}}^1, \{\!\!\{ (c_{\Phi_0(\boldsymbol{k},j)}^1, d_{j,\boldsymbol{k}_0}) \mid j \in [N] \}\!\!\}, \{\!\!\{ (c_{\Phi_1(\boldsymbol{k},j)}^1, d_{j,\boldsymbol{k}_1}) \mid j \in [N] \}\!\!\}, \{\!\!\{ (c_{\Phi_2(\boldsymbol{k},j)}^1, d_{j,\boldsymbol{k}_2}) \mid j \in [N] \}\!\!\})$. Since in the first multiset $\{\!\!\{ (c_{\Phi_0(\boldsymbol{k},j)}^1, d_{j,\boldsymbol{k}_0}) \mid j \in [N] \}\!\!\}$, the element $(c_{\Phi_0(\boldsymbol{k},j)}^1, d_{j,\boldsymbol{k}_0})$ contains distances $d_{j,\boldsymbol{k}_0}, d_{j,\boldsymbol{k}_1}, d_{j,\boldsymbol{k}_2}$, we can calculate a possible coordinates-pair multiset as $P_{\boldsymbol{k}} = \{\!\!\{ \{x_j^{(1)}, x_j^{(2)}\} \mid j \in [N] \}\!\!\}$. As we discussed before, the elements in $P_{\boldsymbol{k}}$ may only have multiplicity value 1 or 2. So we discuss for two possible situations:

1. All the elements (coordinates pairs) in $P_{\boldsymbol{k}}$ either have multiplicity value 2 or the two coordinates within the pair are the same. This means that we can determine all the real coordinates from $P_{\boldsymbol{k}}$, as we discussed in the proof of 4-WL.

2. There exists some element in $P_{\boldsymbol{k}}$, that have two distinct coordinates within the pair, and have multiplicity value 1. We denote the node corresponding to such element $j_0$. We can uniquely find its relevant color (i.e., $c_{\Phi_0(\boldsymbol{k},j_0)}^1, c_{\Phi_1(\boldsymbol{k},j_0)}^1, c_{\Phi_2(\boldsymbol{k},j_0)}^1$) among the three multisets of $c_{\boldsymbol{k}}^2$'s input, because only one $(c_{\Phi_m(\boldsymbol{k},j)}^1, d_{j,\boldsymbol{k}_m})$ will derive the pair of possible coordinates of $j_0$, which is unique. Now, as we discussed in the proof of 3-FWL, we can reconstruct the whole geometry.

$\qquad\square$

### B.3.4 Proof of Universality for Scalars

We have shown in the preceding three subsections that for each $\mathcal{M}(\theta) \in \text{MinMods}$, there exists a parameter $\theta_0$ such that $\mathcal{M}(\theta_0)$ is complete for distinguishing all geometric graphs that are not congruent. Specifically, $\mathcal{M}(\theta_0)$ generates a unique $K$-dimensional intermediate representation for geometric graphs that are not congruent at the output block. According to Theorem 4.1 in Hordan et al. [2023], by passing the intermediate representation through a MLP, $\mathcal{M}(\theta_0)$ can achieve universality for scalars (note that MLPs are already included in the output block of $\mathcal{M}$).

### B.3.5 Proof of Universality for Vectors

The key to proving universality for vectors of function $f_{\text{out}}^{\text{equiv}}$ is Proposition 10 proposed by Villar et al. [2021]. The goal is to prove that function $f = \text{MLPs}(h_m)$ can approximate all functions that are O(3)-invariant and permutation-invariant with respect to all nodes except the $m$-th node (We denote the corresponding group as $\mathcal{G}_m$).

First of all, note that $h_m$ is $\mathcal{G}_m$-invariant. That's because $h_{\boldsymbol{v}}$ ($\boldsymbol{v} \in V^k$) is invariant under O(3) actions, and $h_m$ is obtained by pooling all the $k$-tuples whose first index is $m$. When permuting nodes except the $m$-th one, the multiset $\{\!\{\boldsymbol{v} \mid \boldsymbol{v}_0 = m, \boldsymbol{v} \in V^k\}\!\}$ does not change. According to Theorem 4.1 in Hordan et al. [2023], to prove that $f_{\text{out}}^{\text{equiv}}$ is universal, we need to show that $h_m$ can distinguish inputs $\mathbf{X} \in \mathbb{R}^{n \times 3}$ on different $\mathcal{G}_m$ orbits.

To see this, recall that $h_m = f_{\text{node}}\big(\{\!\{h_{\boldsymbol{v}}^T \mid \boldsymbol{v}_0 = m\}\!\}\big)$, and as we demonstrated in the first three subsections, there always exists some $h_{\boldsymbol{v}}^T$ in the multiset $\{\!\{h_{\boldsymbol{v}}^T \mid \boldsymbol{v}_0 = m\}\!\}$ that can reconstruct the entire geometry. For example, in the case of 3-F-DisGNN, if the point cloud does not degenerate into a 1-D point cloud, there must exist $\boldsymbol{v}$ where $\boldsymbol{v}_0 = m$ and which contains three nodes that are affinely independent (for arbitrary $m$), which can reconstruct the entire geometry as demonstrated in B.3.2. It is worth noting that this reconstruction is started at the $m$-th node, thus if two point clouds share the same $h_m$, they must be related by a $\mathcal{G}_m$ action.

Finally, it is worth noting that an alternative perspective to understand the role of group $\mathcal{G}_m$ is through the **universality for node-wise representation**, $h_m$. The statement "Two geometric graphs are related by a $\mathcal{G}_m$ action" is equivalent to the statement "the two geometric graphs are congruent, and the $m$-th nodes in the two geometric graphs are related by an autoisomorphism." The ability of models to generate node representations that can separate different $\mathcal{G}_m$ orbits implies their capability to generate universal node-wise representations, which is indeed a more powerful property than global universality.

## B.4 Proof of Theorem 6.5

Delle Rose et al. [2023] is our concurrent work, which demonstrated that geometric 3-round 2-FWL and 1-round 3-FWL can distinguish all non-congruent geometric graphs. We refer the readers to read the proof provided by the work, and in comparison to our previous proof, they further make good use of the global properties, i.e., the multiset of all the tuple representations, to get several desired conclusions. In this subsection, we make use of their findings: Since 3-round 2-F-DisGNN and 1-round 3-F-DisGNN are continuous versions of these methods and can achieve injectiveness with a parameter $\theta_0$, they can also achieve completeness.

It is important to note that the update function of 2-E-DisGNN can be used to implement that of 2-F-DisGNN: When the order is 2, the edge representation $e_{ij}^t$ is simplified as $e_{ij}^t = (e_{ij}, h_{ij})$. As a result, the neighbor component of 2-F-DisGNN $\{\!\{(h_{ik}, h_{kj}) \mid k \in V\}\!\}$ is encompassed in the 1-neighbor component of 2-E-DisGNN $\{\!\{(h_{kj}, e_{ik}^t) \mid k \in V\}\!\}$. Hence, the 3-round 2-E-DisGNN is also complete. When the round number is limited to 1, the update function of 3-E-DisGNN can also implement that of 3-F-DisGNN. This is due to the fact that during the initialization step, each 3-tuple is embedded based on the distance matrix. Consequently, the neighbor component of 3-F-DisGNN, $\{\!\{(h_{mjk}, h_{imk}, h_{ijm}) \mid m \in V\}\!\}$, solely represents the distance matrix of $ijkm$, which can be obtained from any $s$-neighbor component ($s \in [3]$) of 3-E-DisGNN (It should be noted that when the iteration number is not 1, determining whether the update function of 3-E-DisGNN can implement that of 3-F-DisGNN is not trivial). Thus, the E-version DisGNNs are also complete. Moreover, as per our proof in Section B.3.4, they are also universal for scalars.

**Discussion of universality for vectors.** The primary obstacle to proving the universality of $\mathcal{M} \in \text{Minmods}_{\text{ext}}$ for vectors lies in the limited expressive power of individual nodes, which may not reconstruct the whole geometry. The problem cannot be easily solved by concating $h_m$ with the global universal representation $h = \{\!\{h_{\boldsymbol{v}}^T \mid \boldsymbol{v} \in V^k\}\!\}$. Specifically, given $(h_{m1}, h_1)$ and $(h_{m2}, h_2)$ for two $N$-node point clouds, if the two tuples are the same, we know that the two point clouds are related by an E(3) transformation and a permutation of the $N$ nodes. However, we cannot guarantee that the permutation (or there exists another permutation) maps the $m$-th node in the first point cloud to the $m$-th node in the second. For instance, even though two points in a point cloud may not be related by an autoisomorphism, they may still have the same representation after applying $\mathcal{M} \in \text{Minmods}_{\text{ext}}$ (Possible counterexamples may be derived from this situation). As a result, $f$ may not be universal for all $G_m$-invariant functions, and $f_{\text{out}}^{\text{equiv}}$ may not be universal for all vector functions. We leave this as an open question.

## C   Detailed Model Design and Analysis

**Radial basis function.** In $k$-DisGNNs, we use radial basis functions (RBF) $f_e^{\mathrm{rbf}} : \mathbb{R} \to \mathbb{R}^{H_e}$ to expand the distance between two nodes into an $H_e$-dimension vector. This can reduce the number of learnable parameters and additionally provides a helpful inductive bias [Klicpera et al., 2020a]. The appropriate choice of RBF is beneficial, and we use nexpnorm RBF defined as

$$f_e^{\mathrm{rbf}}(e_{ij})[k] = e^{-\beta_k(\exp(-e_{ij})-\mu_k)^2}, \tag{35}$$

where $\beta_k, \mu_k$ are coefficients of the $k^{\mathrm{th}}$ basis. Experiments show that this RBF performs better than others, such as the Bessel function used in Klicpera et al. [2020a], Gasteiger et al. [2021].

**Initialization function.** We realize $f_{\mathrm{init}}$ of $k$-DisGNNs' initialization block in Section 5 as follows

$$f_{\mathrm{init}}(\boldsymbol{v}) = \bigodot_{i\in[k]} f_z^i\big(f_z^{\mathrm{emb}}(z_{v_i})\big) \odot \bigodot_{i,j\in[k],i<j} f_e^{ij}\big(f_e^{\mathrm{rbf}}(e_{v_iv_j})\big), \tag{36}$$

where $\odot$ represents Hadamard product and $\bigodot$ represents Hadamard product over all the elements in the subscript. $f_z^{\mathrm{emb}} : \mathbb{Z}^+ \to \mathbb{R}^{H_z}$ is a learnable embedding function, $f_e^{\mathrm{rbf}} : \mathbb{R}^+ \to \mathbb{R}^{H_e}$ is a radial basis function, and $f_z^i, f_e^{ij}$ are neural networks such as MLPs that maps the embeddings of $z$ and $e$ to the common continuous vector space $\mathbb{R}^K$.

$f_{\mathrm{init}}$ can learn an injective representation for $\boldsymbol{v}$ as long as the embedding dimensions are high enough, just like passing the concatenation of $z_{v_i}$ and $e_{v_iv_j}$ through an MLP. We chose this function form for the better experimental performance.

Note that by this means, tuples with different **equality patterns** [Maron et al., 2018] can be distinguished, i.e., get different representations, without explicitly incorporating the representation of equality pattern. This is because in the context of distance graphs, tuples with different equality patterns will have quite different **distance matrices** (elements are zero at the positions where two nodes are the same) and thus can be captured by $f_{\mathrm{init}}$.

**Injective functions.** We realize all the message passing functions as injective functions to ensure expressiveness. To be specific, we embed all the multisets in Equation (7, 9, 10) and in output function with the injective multiset function proposed in Xu et al. [2018], and use the matrix multiplication methods proposed in Maron et al. [2019] to implement the message passing functions of $k$-F-DisGNN (Equation (8)).

**Inductive bias.** Although theoretically there is no need to distinguish tuples with different equality patterns explicitly, we still do so to incorporate inductive bias into the model for better learning. Specifically, we modify the initialization function and the output function as follows: 1. At the initialization step, we learn embeddings for different equality patterns and incorporate them into the results of Equation (36) through a Hadamard product. 2. At the output step, we separate tuples where all sub-nodes are the same and the others into different multisets. These modifications are both beneficial for training and generalization.

# D   Detailed Experiments

## D.1   Experimental Setup

Table 4: Training settings.

|  | MD17 | revised MD17 | QM9 |
|---|---|---|---|
| Train set size | 1000 | 950 | 110000 |
| Val. set size | 1000 | 50 | 10000 |
| batch size | 2 | 2 | 16 |
| warm-up epochs | 25 | 25 | 5 |
| initial learning rate | 0.001 | 0.001 | 0.0005 |
| decay on plateau patience (epochs) | 15 | 15 | 10 |
| decay on plateau cooldown (epochs) | 15 | 15 | 10 |
| decay on plateau threshold | 0.001 | 0.001 | 0.001 |
| decay on plateau factor | 0.7 | 0.7 | 0.5 |

**Training Setting.** For QM9, we use the mean squared error (MSE) loss for training. For MD17, we use the weighted loss function

$$\mathcal{L}(\boldsymbol{X}, \boldsymbol{z}) = (1-\rho)|f_\theta(\boldsymbol{X}, \boldsymbol{z}) - \hat{t}(\boldsymbol{X}, \boldsymbol{z})| + \frac{\rho}{N}\sum_{i=1}^{N}\sqrt{\sum_{\alpha=1}^{3}(-\frac{\partial f_\theta(\boldsymbol{X}, \boldsymbol{z})}{\partial \boldsymbol{x}_{i\alpha}} - \hat{F}_{i\alpha}(\boldsymbol{X}, \boldsymbol{z}))^2}, \quad (37)$$

where the force ratio $\rho$ is fixed as 0.999 (For revised MD17, we set $\rho$ to 0.99 for several targets since the data quality is better).

We follow the same dataset split as GemNet [Gasteiger et al., 2021]. We optimize all models using Adam [Kingma and Ba, 2014] with exponential decay and plateau decay learning rate schedulers, and also a linear learning rate warm-up. To prevent overfitting, we use early stopping on validation loss and an exponential moving average (EMA) with decay rate 0.99 for model parameters during validation and test. We follow DimeNet's [Klicpera et al., 2020a] setting to calculate $\Delta\epsilon$ by taking $\epsilon_{\text{LUMO}} - \epsilon_{\text{HOMO}}$ and use the atomization energy for $U_0, U, H$ and $G$ on QM9. Experiments are conducted on Nvidia RTX 3090 and Nvidia RTX 4090. Results are average of three runs with different random seeds. Detailed training setting can be referred to Table 4.

**Model hyperparameters.** The key model hyperparameters we coarsely tune are the rbf dimension, hidden dimension, and number of message passing blocks. For rbf dimension, we use 16 for MD17 and 32 for QM9. We choose the number of message passing blocks from $\{4, 5, 6\}$. For hidden dimension, we use 512 for 2-DisGNNs and 320 for 3-DisGNNs. The detailed hyperparameters can be found in our codes.

## D.2   Supplementary Experimental Information

**Discussion of results in MD17.** In our experiments on MD17, most of the state-of-the-art performance we achieve is on force targets, and the loss on energy targets is relatively higher, see Table 1. On force prediction tasks, $k$-DisGNNs rank top 2 on force prediction tasks on average, and outperform the best results by a significant margin on several molecules, such as aspirin and malonaldehyde. However, the results on the energy prediction tasks are relatively lower, with 2-F-DisGNN ranking 2nd and 3-E-DisGNN ranking 4th.

This is due to the fact that we have assigned a quite **high weight** (0.999) to the **force loss** during training, similar to what GemNet[Gasteiger et al., 2021] does. In comparison, TorchMD [Thölke and De Fabritiis, 2021] assigned a weight of 0.8 to the force loss, resulting in better results on energy targets, but not as good results on force targets.

Actually, in molecular simulations, force prediction is a more challenging task. It determines the **accuracy** of molecular simulations and reflects the performance of a model better [Gasteiger et al., 2021] (One can see that the energy performance of different models does not vary much). Therefore, it makes more sense to focus on force prediction. Furthermore, previous researches [Batzner et al., 2022, Christensen and Von Lilienfeld, 2020] have found that models can achieve significantly lower

energy loss on the revised MD17 dataset than on the original MD17 dataset, while holding similar force accuracy on the two datasets. As analyzed in Batzner et al. [2022], this suggests that the **noise floor** on the original MD17 dataset is higher on the energies, indicating that better force prediction results are more meaningful than energy prediction results on the original MD17 datasets.

Table 5: MAE loss on revised MD17. Energy (E) in kcal/mol, force (F) in kcal/mol/Å.

| Target | | MACE | Allegro | 2F-Dis. |
|---|---|---|---|---|
| aspirin | E | 0.0507 | 0.0530 | **0.0465** |
| | F | 0.1522 | 0.1683 | **0.1515** |
| azobenz. | E | **0.0277** | **0.0277** | 0.0315 |
| | F | 0.0692 | **0.0600** | 0.1121 |
| benzene | E | 0.0092 | 0.0069 | **0.0013** |
| | F | 0.0069 | **0.0046** | 0.0085 |
| ethanol | E | 0.0092 | 0.0092 | **0.0065** |
| | F | 0.0484 | 0.0484 | **0.0379** |
| malonal. | E | 0.0184 | 0.0138 | **0.0129** |
| | F | 0.0945 | 0.0830 | **0.0782** |
| napthal. | E | 0.0115 | **0.0046** | 0.0103 |
| | F | 0.0369 | **0.0208** | 0.0478 |
| paracet. | E | **0.0300** | 0.0346 | 0.0310 |
| | F | **0.1107** | 0.1130 | 0.1178 |
| salicyl. | E | 0.0208 | 0.0208 | **0.0174** |
| | F | 0.0715 | **0.0669** | 0.0860 |
| toluene | E | 0.0115 | 0.0092 | **0.0051** |
| | F | 0.0346 | 0.0415 | **0.0284** |
| uracil | E | **0.0115** | 0.0138 | 0.0131 |
| | F | 0.0484 | **0.0415** | 0.0828 |

**Revised MD17.** For a comprehensive comparison, we also conducted experiments on the revised MD17 dataset, which has **higher data quality** than original MD17, and compared with two state-of-the-art models, MACE [Batatia et al., 2022] and Allegro [Musaelian et al., 2023]. The results are shown in Table 5. 2-F-DisGNN achieved 10 best and 4 the second best results out of 20 targets. Note that MACE and Allegro are both models that leverage complex equivariant representations. This further demonstrates the high potential of distance-based models in geometry learning and molecular dynamic simulation.

**Supplementary Results on QM9.** We present the full results on QM9 in Table 6. We compare our model with 7 other models, including those that use invariant geometric features: SchNet [Schütt et al., 2018], DimeNet [Klicpera et al., 2020a], DimeNet++[Klicpera et al., 2020b], a model that uses group irreducible representations: Cormorant[Anderson et al., 2019], those that use first-order equivariant representations: TorchMD [Thölke and De Fabritiis, 2021], PaiNN [Schütt et al., 2021] and a model that uses local frame methods: GNN-LF [Wang and Zhang, 2022]. We calculate the average improvements of all the models relative to DimeNet and list them in the table.

Table 6: MAE loss on QM9.

| Target | Unit | SchNet | Cormor. | DimeNet++ | Torchmd | PaiNN | GNN-LF | DimeNet | 2F-Dis. |
|---|---|---|---|---|---|---|---|---|---|
| $\mu$ | D | 0.033 | 0.038 | 0.0297 | **0.002** | 0.012 | 0.013 | 0.0286 | 0.0100 |
| $\alpha$ | $a_0^3$ | 0.235 | 0.085 | 0.0435 | **0.01** | 0.045 | 0.0353 | 0.0469 | 0.0431 |
| $\epsilon_{HOMO}$ | meV | 41 | 34 | 24.6 | **21.2** | 27.6 | 23.5 | 27.8 | 21.81 |
| $\epsilon_{LUMO}$ | meV | 34 | 38 | 19.5 | 17.8 | 20.4 | **17** | 19.7 | 21.22 |
| $\Delta\epsilon$ | meV | 63 | 61 | **32.6** | 38 | 45.7 | 37.1 | 34.8 | 31.3 |
| $\langle R^2 \rangle$ | $a_0^2$ | 0.073 | 0.961 | 0.331 | **0.015** | 0.066 | 0.037 | 0.331 | 0.0299 |
| ZPVE | meV | 1.7 | 2.027 | 1.21 | 2.12 | 1.28 | **1.19** | 1.29 | 1.26 |
| $U_0$ | meV | 14 | 22 | 6.32 | 6.24 | 5.85 | **5.3** | 8.02 | 7.33 |
| $U$ | meV | 19 | 21 | 6.28 | 6.3 | 5.83 | **5.24** | 7.89 | 7.37 |
| $H$ | meV | 14 | 21 | 6.53 | 6.48 | 5.98 | **5.48** | 8.11 | 7.36 |
| $G$ | meV | 14 | 20 | 7.56 | 7.64 | 7.35 | **6.84** | 8.98 | 8.56 |
| $c_v$ | cal/mol/K | 0.033 | 0.026 | 0.023 | 0.026 | 0.024 | **0.022** | 0.0249 | 0.0233 |
| Avg improv. | | -78.99% | -98.22% | 9.42% | 25.00% | 17.50% | 27.82% | 0.00% | 18.83% |
| Rank | | 7 | 8 | 5 | 2 | 4 | 1 | 6 | 3 |

Although our models exhibit significant improvements on the MD17 datasets, their performance on the QM9 datasets is relatively less remarkable. This discrepancy may be attributed to the fact that the QM9 datasets pose a greater challenge in terms of a model's generalization performance: Unlike MD17, QM9 comprises various kinds of molecules and regression targets. However, since $k$-DisGNNs rely purely on the fundamental geometric feature of distance, they may inherently require a larger amount of data to learn the significant geometric patterns within geometric structures and enhance their generalization performance, compared to models that explicitly incorporate equivariant representations or pre-calculated high-order geometric features. Nevertheless, we believe that such learning paradigm, which *employs highly expressive models to extract information from the most basic components in a data-driven manner*, has immense performance potential and will exhibit even greater performance gains when provided with more data points, increased model scales, better data quality or advanced pre-training methods, as already validated in fields such as natural language processing [Brown et al., 2020, Devlin et al., 2018], computer vision [Tolstikhin et al., 2021, Liu et al., 2021, Dosovitskiy et al., 2020], and molecule-related pre-training methods [Xia et al., 2023, Lu et al., 2023].

### D.3 Time and Memory Consumption

We performed experiments on the MD17 dataset to compare the training and inference time, as well as the GPU memory consumption of our models with their counterparts DimeNet and GemNet. The experiments are conducted on Nvidia RTX 3090, and the results are shown in Table 7.

Table 7: Training time, inference time and GPU memory consumption on MD17. The batch size is 32 for 2-F-DisGNN, DimeNet and TorchMD, and 12 for 3-E-DisGNN and GemNet. Training time in ms, inference time in ms, inference GPU memory consumption in MB. Evaluated on Nvidia A100.

|      | 2FDis. | DimeNet | TorchMD | 3EDis. | GemNet |
|------|--------|---------|---------|--------|--------|
| Asp. | 234/66/2985 | 270/74/5691 | 101/39/2072 | 873/273/18585 | 556/182/7169 |
| Ben. | 92/24/1058 | 162/44/1815 | 110/41/905 | 224/61/3644 | 380/130/1789 |
| Eth. | 61/18/637 | 163/43/791 | 103/43/532 | 112/30/1561 | 462/158/804 |
| Mal. | 61/18/640 | 161/43/791 | 103/42/532 | 113/30/1561 | 356/118/804 |
| Nap. | 184/50/2193 | 238/68/4578 | 111/45/1692 | 652/185/11804 | 444/141/5253 |
| Sal. | 137/37/1758 | 206/58/3468 | 119/42/1401 | 458/127/8281 | 405/136/3778 |
| Tol. | 131/36/1560 | 200/54/3126 | 107/25/1326 | 399/109/6915 | 473/153/3328 |
| Ura. | 93/24/1058 | 169/45/1782 | 113/42/906 | 225/61/3644 | 369/136/1776 |
| Avg. | 124/34/1486 | 196/53/2755 | 108/39/1170 | 382/109/6999 | 430/144/3087 |

2-F-DisGNN demonstrates significantly better time and memory efficiency compared to DimeNet and GemNet. Even when compared to the quite efficient model, TorchMD, 2-F-DisGNN shows competitive efficiency. This is partly because it utilizes high-order tensors and employs dense calculation, resulting in significant acceleration on GPUs. However, due to the high theoretical complexity and high hidden dimension, 3-E-DisGNN shows relatively worse performance. Regarding this limitation, we have listed several possible solutions in Section 8 and recent work on simplifying and accelerating $k$-WL [Zhao et al., 2022, Morris et al., 2022] when $k$ is large may also be applied, which is left for future work. Nonetheless, given that 2-F-DisGNN already satisfies the requirements for both theoretical (Section 6.3) and experimental (Section 7) performance, our primary focus lies on this model, and the analysis further emphasizes the practical potential of 2-F-DisGNN.

## E Supplementary Related Work

There is a wealth of research in the field of interatomic potentials, which introduced various methods for modeling the atomic environment as representations [Bartók et al., 2010, 2013, Shapeev, 2016, Behler and Parrinello, 2007]. In particular, ACE (Atomic Cluster Expansion) [Drautz, 2019] has presented a framework that serves as a unifying approach for these methods and can calculate high-order complete polynomial basis features with small cost regardless of the body order. Building on Drautz [2019], Batatia et al. [2022] proposed a GNN variant called MACE that can effectively exploit the rich geometric information in atoms' local environment. Additionally, Joshi et al. [2023] proposed

geometric variants of the WL test to characterize the expressiveness of invariant and equivariant geometric GNNs for the general graph setting. All of Drautz [2019], Batatia et al. [2022], Joshi et al. [2023] leverage the many-body expansion and can include $k$-tuples during embedding atom's neighbors, allowing atoms to obtain $k$-order geometric information from their local environments like $k$-DisGNNs.

However, we note that there are key differences between these works (based on many-body expansion) and $k$-DisGNNs in terms of their message passing units and use of representations. The former assumes that the total energy can be approximated by the sum of energies of atomic environments of individual atoms and therefore use atoms as their message passing units, while $k$-DisGNNs pass messages among $k$-tuples and pool tuple representations to obtain global representation. In addition, while interatomic potential methods leverage equivariant representations to enhance their expressiveness, $k$-DisGNNs completely decouple E(3)-symmetry and only use the most basic geometric invariant feature, distance, to achieve universality and good experimental results.

