# OpenReview forum: "Is Distance Matrix Enough for Geometric Deep Learning?"
_NeurIPS.cc/2023/Conference — NeurIPS 2023 poster_

### Official Review · Reviewer_fGWC · 2023-07-04

**Soundness:** 3 good
**Presentation:** 3 good
**Contribution:** 2 fair
**Rating:** 5
**Confidence:** 4

**Summary:**

This paper proposes k-DisGNNs, that can use distances alone to represent higher-order interactions, instead of the direct use of angles, torsional angles, and dihedral angles. The authors try to build a bridge between geometric deep learning and the traditional expressiveness of graph representation learning without geometric information. They conduct experiments mainly on MD17 and show results on QM9 in the appendix. For MD17, 2F-Dis. and 3E-Dis. are comparable and SOTA, but for QM9 (still small geometric graphs), 2F-Dis. is worse than several previous works and 3E-Dis. can not be implemented due to high complexity, which means their method with higher k cannot extend to larger geometric graphs with more nodes.

**Strengths:**

1. The first work to use distances alone to represent higher order geometric information including angles, torsional angles, and dihedral angles.
2. They establish a connection between geometric deep learning (GDL) and traditional graph representation learning (GRL).
3. Good performances of k=2 model on MD17.
4. Figures are good and informative.

**Weaknesses:**

1. Prohibitive Computational Costs for k > 2: The computational demands associated with the proposed method escalate dramatically when k > 2, as demonstrated in the appendix. Specifically, when k is set to 3, the model becomes computationally infeasible for the QM9 dataset, which comprises small molecules.

2. Lack of Scalability and Performance for k = 2: When applied to the modestly-sized MD17 dataset, the method exhibits satisfactory performance with k = 2. However, this performance deteriorates markedly when transitioning to the more extensive QM9 dataset. The method's performance, as evaluated with 2F-Dis, fails to measure up to that of prior approaches such as TorchMD, GNN-LF, and PaiNN.

3. The writing about the message passing mechanism lacks clarity and is hard to follow.

Major Concern: The major limitation of the proposed method is its inability to maintain the good performances when scaling to larger datasets, particularly QM9 concerning small molecules with not so many atoms. This significantly constrains the method's applicability and undermines its contribution.

**Questions:**

As listed above in weaknesses.

**Limitations:**

They mentioned the limitations in the main paper. More potential limitations are listed above.

---

> ### Author Rebuttal · Authors · 2023-08-09
>
> We thank Reviewer fGWC for the careful and detailed review. We address the comments below.
>
> > W1: Prohibitive Computational Costs for k > 2: The computational demands ... escalate dramatically when k > 2, ..., when k is set to 3, the model becomes computationally infeasible for the QM9 dataset, ...
> >
>
> Thanks for raising the concern. To respond, we re-measured the computational cost of all models presented in Table 7 of our paper, and added an additional efficient and high-performing model, TorchMD$^{[1]}$, for comparison. The new results are shown in Table 1 of the PDF (see global rebuttal). Additionally, we tested the dense version of Dime/GemNet for reference (please see Weakness 2 raised by Reviewer UsYi and our response).
>
> 1. We first recognize that the memory usage does increase dramatically due to the global message passing for k > 2. Nevertheless, we have already **discussed potential solutions** in both Sec. 8 and Appendix D.4 of our paper. Also, please note that 3-E-DisGNN is even **faster than** its counterpart GemNet on MD17.
> 2. We clarify that 3-E-DisGNN is still **computationally feasible** for QM9, although taking longer to train and tune due to the larger amount of training data points. In Fig. 1 of the PDF, we show the train/validation loss curves of 3-E-DisGNN on QM9 in the first tens of epochs, which are converging normally and approaching the validation accuracy of 2-F-DisGNN, thus validating the feasibility. Please note that 2-F-DisGNN is already a complete model on QM9, therefore we mainly focus on its performance.
> 3. We finally would like to emphasize that **the primary model we propose is 2-F-DisGNN**, which is proven to be universal for scalar functions (the most cases concerned) and exhibited great experimental performance. As shown in Table 1, 2-F-DisGNN is \~2x more efficient (both in speed and memory usage) compared with DimeNet and 4x\~7x more efficient compared with GemNet on md17, while having significantly better prediction results. It is competitive even when compared to the efficient model TorchMD.
>
> > W2: Lack of Scalability and Performance for k = 2: ... performance deteriorates markedly when transitioning to the more extensive QM9 dataset ... method's performance ... fails to measure up to that of prior approaches such as TorchMD ...
> >
>
> We first clarify that the performance of 2-F-DisGNN does not “deteriorate markedly” on QM9: As shown in Table 6 in the appendix, 2-F-DisGNN is highly competitive with the baselines, with 14.27% gain over DimeNet compared to the SOTA models TorchMD (25%) and GNN-LF (27.82%).
>
> Nevertheless, we acknowledge that we have spent less time tuning 2-F-DisGNN on QM9 compared to MD17. Therefore, we further tuned it during the rebuttal period. The results are now presented in Table 2 of the new PDF.
>
> We believe a data-driven model relying on pure distance needs a bit more layers to extract the various geometric information from more extensive dataset like QM9, as we discussed in Appendix D.3 of the paper. So what we have added to the model is just several residual layers in interaction or output blocks. Note that this addition does minor effect to the efficiency of the model.
>
> As shown in Table 2, though we have limited time to perform tuning and only tuned several targets, the model gains a **consistent improvement** on all targets we tuned, especially on $\epsilon_{HOMO}$, $\epsilon_{LUMO}$ and $\Delta \epsilon$. On $\Delta \epsilon$, it even achieves a **new SOTA** result of 31.96. Also please note that target $U$ to $G$ share similar property with $U_0$, thus will also  We believe with further tuning, maybe by stacking more residual layers to extract the intermediate geometric reprentations (as is commonly done in chemical/physical models such as Phys$^{[2]}$/Dime/GemNet) or applying pre-training methods$^{[3,4]}$, one could further release the potential of DisGNNs. Additional discussion regarding potential methods to enhance performance can be seen in Appendix D.3 of our paper.
>
> Moreover, we also conduct some preliminary experiments on 3BPA during rebuttal, a dataset testing the performance of models on out-of-domain data (See Sec. 5.3.2 in [5] for more details). We compared our models to SOTA models shown in [5], and the results are shown in Table 3 in the PDF. As one can see, our model shows competitive or better performance compared to the other models using complicated equivariant representations. Additional improvements could be achieved by further tuning (we only tuned for two rounds due to the time limit).
>
> Finally, we emphasize that DisGNNs are purely data-driven models based on distance matrices, without using any domain-specifc knowledge like the angles in DimeNet and dihedral angles in GemNet. We believe the potential of such a highly-expressive data-driven model is significant, as already validated in fields like NLP, CV and molecule pretraining methods$^{[3,4]}$. We will continue to improve and evaluate our models. Thanks again for the insightful comment!
>
> > W3: The writing about the message passing mechanism lacks clarity and is hard to follow.
> >
>
> Thanks for the valuable comment. Due to the page limit, the formulaic details of the message passing blocks are included in Appendix C, and we will add a hyperlink in the main body for better presentation.
>
> For the Major Concern, Please see our response to W1 and W2.
>
> [1] Thölke, et al. "Torchmd-net: equivariant transformers for neural network based molecular potentials." (2022).
>
> [2] Unke, et al. "PhysNet: A neural network for predicting energies, forces, dipole moments, and partial charges." (2019).
>
> [3] Zhou, Gengmo, et al. "Uni-Mol: a universal 3D molecular representation learning framework." (2023).
>
> [4] Zaidi, Sheheryar, et al. "Pre-training via denoising for molecular property prediction." (2022).
>
> [5] Batatia, et al. "MACE: Higher order equivariant message passing neural networks for fast and accurate force fields." (2022).

---

> > ### Comment · Reviewer_fGWC · 2023-08-10
> > **Responses to Rebuttal**
> >
> > My concerns about scaling to QM9, model performances, and performance gains beyond previous works are partially solved. But it seems to me for the QM9 and the other dataset, the proposed method cannot beat state-of-the-art methods, for example TorchMD and MACE.
> >
> > Hence, I am willing to adjust my score to 5.

---

> > > ### Author Response · Authors · 2023-08-13
> > > **Response to Reviewer fGWC**
> > >
> > > Thank you for the careful reading and prompt feedback. We are glad that your concerns have been partially solved, and we will continue to improve our model's experimental performance on these datasets.
> > >
> > > We notice that the original score has not been updated yet. Could you please increase the score by editing the original review?

---

### Official Review · Reviewer_5v5F · 2023-07-06

**Soundness:** 3 good
**Presentation:** 3 good
**Contribution:** 3 good
**Rating:** 7
**Confidence:** 4

**Summary:**

The paper proposes $k$-DistGNNs, a sequence of k-(F)WL-like models that operate on the distance matrix of a geometric graphs. These models are able to compute higher-order geometric invariants, generalise existent models, and for certain values of $k$ are shown to be able to approximate any geometrically invariant or equivariant functions. The model is evaluated against other SOTA models on standard molecular benchmarks. Additionally, the paper proposes a procedure to generate new counter-examples of geometric graphs that cannot be distinguished by MPNNs operating on the distance matrix.

**Strengths:**

-	The paper is very well written and easy to follow.
-	The expressive power of Geometric GNNs is a relatively new and unstudied topic, and the paper makes an important contribution in this regard.
-	The proposed approach is a simple, but powerful, extension of k-WL to distance matrices and implicitly geometric graphs.
-	The evaluation shows that the proposed models perform very well on standard benchmarks against the latest geometric models (outperforming them in some tasks)
-	The universal approximation properties of the model are very attractive and back up the claim that the distance matrix is “enough”. Although, the question in the title is somewhat rhetorical since the distance matrix fully determines geometric graphs. So just from that, it follows the information contained inside is enough.
-	The new counterexamples that the paper provides are very welcome as well as the general procedure used to generate them.


**Weaknesses:**

- Note that WL generalisations for Geometric GNNs were also studied in _On the Expressive Power of Geometric Graph Neural Networks_ (ICML 2023), which is not mentioned in the text as far as I could see.

**Questions:**

N/A

**Limitations:**

Yes.

---

> ### Author Rebuttal · Authors · 2023-08-09
>
> We thank Reviewer 5v5F for the careful review. We address the comments below.
>
> > Note that WL generalisations for Geometric GNNs were also studied in On the Expressive Power of Geometric Graph Neural Networks (ICML 2023), which is not mentioned in the text as far as I could see.
> >
>
> Thanks for the reminder. Indeed, we have noticed the great work "On the Expressive Power...", which offers a comprehensive analysis of the expressive abilities of geometric graph neural networks. In Appendix E of our paper, we briefly discussed the similarities and differences between our work and theirs, and will further discuss the work in detail in the revised version.
>
> We hope our response can address the problems, and please let us know if you have additional questions.

---

### Official Review · Reviewer_w5WJ · 2023-07-06

**Soundness:** 4 excellent
**Presentation:** 3 good
**Contribution:** 3 good
**Rating:** 7
**Confidence:** 3

**Summary:**

In this paper, the authors argue that the distance matrix associated with a geometric graph is sufficient to build powerful geometric neural networks.
In particular, the paper proposes a new family of graph architectures, named k-DisGNN, operating on a complete graph with node distances on the edges.
These architectures, inspired by the k-WL test (and its variations like the k-FWL or k-EWL tests), are able to model higher-order geometric information, which makes them more expressive than vanilla message-passing networks and ensures their universal approximator property.


**Strengths:**

The paper is well written and well motivated.
The proposed ideas lead to a relatively simple yet expressive design to process geometric graphs.
The experiments support the theoretical results in the main paper.


In addition, the supplementary materials include a number of examples where message-passing networks fail to distinguish geometric graphs.
These interpretable examples can be useful to understand the limitations of existing and future architectures.


**Weaknesses:**


Some details about the proposed models were not completely clear to me, although this might be due to my little familiarity with Weisfeiler-Leman algorithms.
Still, I think the presentation could be made a bit clearer, see Questions section.

I don't see major flaws in this paper.


**Questions:**

I am not sure I understand the difference between Eq. 4 and 5.
It seems to me that the indices $j$ and $w$ are just swapped, but in the end the set contains the same tuples.
Could you clarify this?


lines 197-203: I think that writing the message passing block explicitly would improve the clarity of the paper (rather than explaining it by discussing how it differs from Eq. 4 and 5).
Similarly, in line 216, it would be useful to explicitly include the message passing equation of k-E-DisGNN.


In Sec. 5 k-DisGNN and k-F-DisGNN refer to two different models (using respectively the k-WL and k-FWL paradigms for message passing) but in Sec. 7 k-DisGNN seems to be used to refer to both 2-F-DisGNN and 3-E-DisGNN in an abstract way.
This seems confusing.


Are you results valid only for 3D data or do these results generalize to any dimension?


**Limitations:**

The authors explicitly addressed the limitations of the work in the main paper.

---

> ### Author Rebuttal · Authors · 2023-08-09
>
> We thank Reviewer w5WJ for the insightful and detailed comments. We address the comments below.
>
> > I am not sure I understand the difference between Eq. 4 and 5. It seems to me that the indices j and w are just swapped, but in the end the set contains the same tuples. Could you clarify this?
>
> Thanks for the question. It is correct that WL and FWL process identical tuples, but they are different in their ways of grouping these tuples. FWL groups tuples in a more compact way: It first groups all tuples associated with a given point $j$ (Eq. 3), then proceeds to iterate through such points (Eq. 5) in a multiset. WL first groups all tuples associated with the $j$-th position (Eq. 2), then enumerates the position (Eq. 4). Notably, while it's possible to derive WL's input from that of FWL, the reverse process is not feasible. This asymmetry highlights the expressiveness advantage of FWL over WL methods intuitively.
>
> > I think that writing the message passing block explicitly would improve the clarity of the paper (rather than explaining it by discussing how it differs from Eq. 4 and 5). Similarly, in line 216, it would be useful to explicitly include the message passing equation of k-E-DisGNN.
>
> We thank the valuable suggestions from the reviewer. Due to the page limit, the details of the message passing blocks are included in Appendix C. We will add a hyperlink to it in the main paper.
>
> > in Sec. 7, k-DisGNN seems to be used to refer to both 2-F-DisGNN and 3-E-DisGNN in an abstract way. This seems confusing.
>
> We thank the reminder from the reviewer. Indeed, we define k-DisGNN**s** (with an additional "s" at the end) as a general term for various versions of k-DisGNN models, see the first paragraph in Sec. 5. Throughout Sec. 7, we continue to refer to these models as "k-DisGNNs" by a general name. We will highlight this definition in the text.
>
> > Are you results valid only for 3D data or do these results generalize to any dimension?
>
> The theoretical results, specifically theorems 6.4 and 6.5, can generalize to high-dimension situations since $n$D Euclidean space ($n > 3$) shares similar topological properties as 3D space. For example, in $n$D space, one point’s position can be determined to at most two possible candidates if given the distances between the point to n affinely independent points. And these two candidates are mirrorly symmetric w.r.t. the hyper-plane formed by the n points, which is a strong geometric constraint in $n$D space -- similar to the one we repeatedly use in the proof of 3D situations. Thus one could, for example, similarly prove that 3-round $(n-1)$-E/F-DisGNN or 1-round $n$-E/F-DisGNN are universal scalar function approximators over geometric graphs in $n$D space. However, we mainly focus on 3D situations because the real-world structures of most concerns, such as molecules or general point clouds, are 3D data. Nevertheless, we all agree on the importance of generalizing the results to the $n$D situation, which we will refine in our future work.
>
>
>
> Thanks again for the comments, and we hope our response can address the problems.

---

> > ### Comment · Reviewer_w5WJ · 2023-08-15
> >
> > I thank the authors for the complete answers.
> >
> > I would still recommend the authors to include the explicit equations in the main paper in the camera ready version.
> > Moreover, while generalising all the results to n>3 dimensions can be a future work, I think the author's answer is already an interesting result which should be briefly mentioned in the paper (maybe in the appendix?).
> >
> > I maintain my acceptance recommendation.

---

> > > ### Author Response · Authors · 2023-08-17
> > > **Response to Reviewer w5WJ**
> > >
> > > We express our gratitude to the reviewer for the valuable suggestions and feedback. We will consistently refine our statements and discuss more about these points in our revised version.

---

### Official Review · Reviewer_UsYi · 2023-07-08

**Soundness:** 3 good
**Presentation:** 2 fair
**Contribution:** 3 good
**Rating:** 7
**Confidence:** 4

**Summary:**

This work proposes neural networks for operating on distance matrices obtained from 3D/Euclidean point clouds. New architectures which generalise the k-WL/k-tuple higher order GNN framework (Morris et al.) to distance matrices are proposed, along with proofs of their universality/completeness for k=2 (scalar) and k=3 (vector predictions). The approach is evaluated to work as well or better than current geometric GNNs for small molecular tasks like MD17 and QM9.

**Strengths:**

- This work presents a **novel** k-WL framework for complete distance graphs from Euclidean point clouds. New architectures for processing k-tuple distance graphs are proposed, in a spirit similar to influential work by Morris et al. (https://arxiv.org/abs/1810.02244) on higher order GNNs for standard graphs.

- I believe the theoretical results showing that architectures based on 2-WL and 3-WL variants of the proposed test are complete for scalar/vector predictions on distance graphs is **significant**.

- The proposed architectures are shown to **work as well or better** than existing geometric GNNs for small molecular simulation tasks (QM9, MD17, rMD17).

- The procedures to construct families of counterexamples for 1-EWL and vanilla disGNNs are described in detail and seem interesting.

**Weaknesses:**

- I don't have major concerns.

- Ambiguous claim re. Geometric Deep Learning:
    - Geometric Deep Learning is a pretty broad field beyond point clouds embedded in Euclidean space. It includes Groups, Gauges, Meshes, Grids, and Graphs. I personally felt the title and claims within the paper could be perceived ambiguously because the findings of this paper are not directly applicable across **the entire** spectrum covered by GDL; it only covers geometric *graph* learning.
    - Is this work relevant to other areas of GDL? If so, how?

- Scalability: One major downside of the proposed approach of adapting higher order tuple-based GNNs to operate on distance graphs will be the computational cost of working with anything other than small molecules.
    - On MD17/rMD17, the models use a batch size of 2. I suppose this is due to high memory usage.
    - Regarding the experiments on timings in the appendix: this seems like an unfair comparison b/c one would naturally expect using dense tensor-based implementations like yours to be much faster than PyG implementations of DimeNet/GemNet (especially for small molecule graphs with tens of nodes).
    - **Is it possible to compare apples-to-apples the timing/memory usage of dense implementations of DimeNet/GemNet, too?**
    - I see this paper's main contribution to be theoretical, so I don't think this is a major concern.

- I had difficulty following Proof B.3.1. Perhaps proving a proof sketch or some warmup text outlining the proof techniques may be helpful, especially as it spans several pages.
    - (It may be that I need more time to study the proof in further detail.)

**Questions:**

Questions and clarifications:

- In Related Work, for Hordan-etal, 2023: I realise this is probably concurrent work, but isn't the Gram matrix and distance matrix interchangable?

- In Section 4, for the family of counterexamples for 1-EWL and vanilla disGNNs: how do you see these counterexamples being useful or informing future research? (Beyond telling us what we already know from the counterexample of Pozdnyakov-Ceriotti.)

Suggestions:

- Lines 85-88: I would suggest stating that these results are for complete graphs.

- Lines 130-135: It was unclear what orientation information meant in this context without a coordinate system. Orientation w.r.t. what?

- Line 1293 - typo.

- I would personally go with Geometric Graph Learning instead of GDL. I realise this is opinionated.

**Limitations:**

The authors have adequately addressed technical limitations but have not stated potential negative social impact.

Beyond what the authors mention regarding scalability of their models, one major theoretical limitation of the proposed framework is that it is restricted to full geometric graphs, and the construction of complete/universal models for the general sparse graph setting remains an open question. This my be worth reiterating.

---

> ### Author Rebuttal · Authors · 2023-08-09
>
> We thank the Reviewer UsYi for their detailed and insightful review. We address the comments below.
>
> > W1: Ambiguous claim re. Geometric Deep Learning
>
> Thanks for the insightful comment. Indeed, in this paper, we mainly use the term geometric deep learning to refer to learning over points embedded in 3D Euclidean space, such as general point clouds and molecules, but do not cover the more broad types of non-Euclidean data such as manifolds and meshes. We will make it clear in the revision. Beyond that, we also notice that the most important applications of GDL lie in networks, graphics, particle physics, molecules, etc., which can all be represented by points with some coordinates and/or local affinity defined, thus may be solved by our methods. Therefore, we expect our k-DisGNNs to become a general technique for a wide range of GDL problems beyond the problems considered in the paper.
>
> > W2-1: … to compare apples-to-apples … with dense … DimeNet/GemNet.
>
> We agree that it’s necessary to compare their performance and efficiency apples-to-apples. We’ve implemented a dense-version DimeNet/GemNet by replacing all the sparse operations (mainly “scatter”) with dense matrix multiplications and tested their performance. The results are shown in Table 1 in the PDF (in global rebuttal). As can be observed, the dense operation **does not bring obvious performance boost for the two models**. This is expected, since the graphs they process have relatively high sparsity. And GemNet’s official code has already been optimized to be much efficient. Thus its performance did not change much after we rewrote it.
>
> The results show that our universal (scalar) model 2F-DisGNN is more efficient compared to the two models and also achieves much better performance. Please refer to our response to W1 raised by Reviewer fGWC for more analysis of Table 1.
>
> > W2-2: … I suppose this is due to high memory usage.
>
> We clarify that it is not due to memory usage, but rather for improved generalization performance. This approach is similar to what is mentioned in Appendix E of GemNet.
>
> > W3: I had difficulty following Proof B.3.1. Perhaps proving a proof sketch … may be helpful …
>
> Thanks for the valuable suggestion, and we will include a proof sketch in our revised paper for better elaboration. Due to words limit of author rebuttal, we would like to provide the proof sketch in discussion period.
>
> > Q1: … isn't the Gram matrix and distance matrix interchangable?
>
> While they are interchangeable when they represent the entire graph, this interchangeability diminishes when they are **constrained to represent local tuples**. For example, if one takes the Gram matrix of a 3-tuple ijk as its initial representation (as done in Appendix B.2. in Hordan-etal), actually 6 distances are encompassed: 3 distances among ijk, and additional 3 distances from ijk to the center c. However if one takes the distance matrix of ijk (as we done), only the distances among ijk are given. So it could be more challenging to recover the whole geometry from distance matrix, **and it can be more “clean” and significant**: pair-wise distance can provide more **inductive bias** for modeling interaction intensity or relationship between nodes (compared to pair-wise angles w.r.t. center).
>
> Furthermore, it is noteworthy that Hordan-etal's primary model, as detailed in their main body, leverages coordinate projection more than Gram matrix (see Eq. 3 of Hordan-etal's). This approach, while potentially more straightforward, could negatively impact the inductive bias mentioned previously, thereby may limit its real-world applicability (see Table 2 of Hordan-etal 2023).
>
> > Q2: … how do you see these counterexamples being useful or informing future research?
>
> By understanding the symmetry contained in the counterexamples, researchers can design more powerful geometric models by breaking such symmetry. This is in spirit similar to those works in traditional GRL, such as subgraph GNNs$^{[1]}$: For example, by just embedding the 1-hop geometric subgraph around each node in Fig.1 in our paper, Vanilla DisGNN can easily distinguish the two geometric graphs. This approach could inspire complete but more efficient models. We are exploring it in our future work.
>
> > Suggestion: … Orientation w.r.t. what?
>
> We clarify that what we mean in line 128 is, by combining the universal scalar with coordinates in a specific coordinate system (what orientation means here), powerful equivariant values can be produced. We do not suggest that orientation can be represented without a coordinate system. However, coordinate systems here only provide a reference to characterize the orientation of equivariant targets, and only appear at the output head of models, resulting in a cleaner model design. Examples include Theorem 6.2’s third property and calculating the derivative of molecule’s energy w.r.t. atoms’ coordinates to obtain force vectors acting on atoms.
>
> And thanks a lot for the other suggestions! We will continue to refine our statements.
>
> > Limit: One major theoretical limitation …
>
> We thank the insightful comment from the reviewer; it's indeed an interesting topic for discussion. Our opinion on this point inclines towards a positive outlook: distance matrix itself can be somehow abundant, since it contains n*n elements while only corresponds to n individual nodes. Appropriately sparsifying it may not influence the geometric structure represented by it, thus may not influence the theoretical completeness. An intuitive understanding is: a point’s position can be determined with at most 4 distances (to 4 affinely independent points), so the remaining n-5 distances might be deemed redundant. It is an intriguing subject to study how to appropriately sparsify it while maintaining the good experimental results. We will explore it in our future work.
>
> Thanks again for the comments! We hope our response can address the problems.
>
> [1] Zhang, Muhan, and Pan Li. "Nested graph neural networks."

---

> > ### Comment · Reviewer_UsYi · 2023-08-14
> >
> > > Beyond that, we also notice that the most important applications of GDL lie in networks, graphics, particle physics, molecules, etc., which can all be represented by points with some coordinates and/or local affinity defined, thus may be solved by our methods.
> >
> > I disagree that applications on manifolds and meshes are not 'important applications of GDL'.
> >
> > As you stated in your response, the proposed methods cannot be applied to *all* of GDL -- so the claims around GDL are incorrect.
> >
> > > We clarify that it is not due to memory usage, but rather for improved generalization performance. This approach is similar to what is mentioned in Appendix E of GemNet.
> >
> > I don't understand how using a small batch size of 2 improves generalisation? I re-read Appendix E of GemNet around variance after message passing.
> >
> > > Our opinion on this point inclines towards a positive outlook: distance matrix itself can be somehow abundant, since it contains n*n elements while only corresponds to n individual nodes. Appropriately sparsifying it may not influence the geometric structure represented by it, thus may not influence the theoretical completeness. An intuitive understanding is: a point’s position can be determined with at most 4 distances (to 4 affinely independent points), so the remaining n-5 distances might be deemed redundant. It is an intriguing subject to study how to appropriately sparsify it while maintaining the good experimental results. We will explore it in our future work.
> >
> > Okay, I agree its an interesting mathematical problem regarding whether geometric structure can be maintained while sparsifying an n x n distance matrix. However, the current theoretical framework and models (to my understanding) necessarily require a full n x n matrix and are not directly applicable to the sparse setting. I think this is a valid limitation and would suggest reiterating/discussing it in the revised paper.

---

> > > ### Author Response · Authors · 2023-08-16
> > > **Response to Reviewer UsYi [1/2]**
> > >
> > > We thank the careful reading and prompt feedback from the reviewer. We address the comments below.
> > >
> > > > I disagree that applications on manifolds and meshes are not 'important applications of GDL'.
> > > As you stated in your response, the proposed methods cannot be applied to *all* of GDL -- so the claims around GDL are incorrect.
> > > >
> > >
> > > Thanks for the careful comments regrading this point. We agree that applications on manifolds and meshes are also important, and some of them are included in "graphics" in our previous response. We also expect DisGNNs to become a potential technique for these two domains for the following reasons: 1) Meshes are naturally formed by k-tuples, where our k-DisGNNs might apply. This idea is partly supported by previous work$^{[1, 2]}$, which respectively treat edges (2-tuple) and triangles (3-tuple) as the basic units for convolution, and demonstrate promising results in different tasks. 2) Like Euclidean space is often discretized as grids, 2D manifolds in computer graphics are often discretized as meshes too, where our models might apply. Nevertheless, we acknowledge these are just intuitive directions, and we do not cover these fields in our paper as of now. We will make it more clear in the revision. Thanks again for the feedback!
> > >
> > > > I don't understand how using a small batch size of 2 improves generalisation? I re-read Appendix E of GemNet around variance after message passing.
> > > >
> > >
> > > We apologize for the misrepresentation; it’s indeed mentioned in Appendix G of GemNet$^{[3]}$: “We found the selection of the batch size to be of great influence on the model’s performance for the MD17(@CCSD) dataset, (thus set batch size to 1 for GemNet on MD17 and MD@CCSD)”.
> > >
> > > There have been researches$^{[4, 5]}$ on the effect of batch size, supporting the idea that smaller batch sizes can have a regularization effect due to the inherent noise in the gradient estimation. Like GemNet, we found a batch size of 2 to be the optimal choice for our models, and further decreasing the batch size (to 1) does not lead to improved generalization performance.
> > >
> > > > … However, the current … models … necessarily require a full n x n matrix and are not directly applicable to the sparse setting. I think this is a valid limitation and would suggest reiterating/discussing it in the revised paper.
> > > >
> > >
> > > We thank the suggestion from the reviewer, and we will discuss more about this in the revised paper. We clarify several points regarding this.
> > >
> > > 1. We agree that this is a limitation of the work as of now. However, we do not think the complete distance matrix is a necessary condition for our models to achieve completeness: as we respond in previous rebuttal and the subsequent point, we hold a positive outlook on this open problem.
> > > 2. Like distance matrices, **k-DisGNNs also have the potential to maintain completeness** while being appropriately sparsified. For example, in the proof of the completeness of 4-DisGNN (see Appendix B.3.1. of our paper), we only use 4 distances from a single point $j$ to the selected 4-tuple to determine the 4 pairs of two candidate positions of $j$ (see line 998 - 1007). The remaining N-5 distances are not required. Therefore, if the distance matrix is appropriately sparsified (i.e., to cut the interactions happening within the N-5 relations), $j$'s position can still be theoretically determined, and the model can still be complete. This is consistent with what we explained in the previous response.
> > > 3. We respectfully disagree that our models “are not directly applicable to the sparse setting”. The potential sparse solutions we provide in Sec. 8 and Appendix D.4 are actually very **easy/direct to implement** for DisGNNs (though they may compromise the completeness but are common ways for modern models to trade-off between theoretical completeness and tractability$^{[3]}$). For example, various cutoffs can be introduced to the model, such as embedding cutoff (to ignore tuples with long “diameter”) and interaction cutoff (to ignore long interactions). For the embedding cutoff, one can first define a metric function $f_{diam}: \mathbb{R}^{(k^2 - k)/2} \to \mathbb{R}$ to calculate the “diameter” of a $k$-tuple from its local distance matrix, and $max$ is an easy example. Therefore, there is in fact a wide design space for sparsifying our model, which we do not touch much in this paper since we mainly focus on theoretical completeness.

---

> > > > ### Author Response · Authors · 2023-08-16
> > > > **Response to Reviewer UsYi [2/2]**
> > > >
> > > > > W3: I had difficulty following Proof B.3.1. Perhaps proving a proof sketch … may be helpful …
> > > > >
> > > >
> > > > We provide the proof sketch as follows. Intuitively, the proof includes mainly four steps:
> > > >
> > > > 1. To obtain the representation of the 4-tuples $\boldsymbol{k}$, where the 4 nodes are affinely independent, from geometric 4-WL’s output. (line 973 - 986)
> > > > 2. To calculate the coordinate candidates (at most two for each node) of other nodes given the representation of $\boldsymbol{k}$. (line 987 - 1014)
> > > > 3. Since one can calculate other nodes’ coordinate candidates from each $C^0_{i, \boldsymbol{k}}, i \in \\{0, 1, 2, 3\\}$ (see line 991), four possible geometry sets (each set has at most $2^{(N-4)}$ possible geometric structures because of the uncertainty of the coordinate candidates) can be obtained. By repeatedly refining the four geometry sets, one can finally get the only possible geometry. Including:
> > > >     1. To calculate all the determinable points within each geometry set and refine all the geometry sets (lines 1015 - 1037)
> > > >     2. To compare the four geometry sets to determine each previously undeterminable point’s coordinates one by one. (line 1038 - 1066)
> > > > 4. To discuss about the corner cases when there is no such 4-tuple where 4 nodes are affinely independent. (line 1067 - 1069)
> > > >
> > > > Thanks again for the careful comments and feedback! We hope our response can address the concerns.
> > > >
> > > > [1] Hanocka, Rana, et al. "Meshcnn: a network with an edge."
> > > >
> > > > [2] Hertz, Amir, et al. "Deep geometric texture synthesis."
> > > >
> > > > [3] Gasteiger, et al. "Gemnet: Universal directional graph neural networks for molecules."
> > > >
> > > > [4] Keskar, et al. "On large-batch training for deep learning: Generalization gap and sharp minima."
> > > >
> > > > [5] Masters, et al. "Revisiting small batch training for deep neural networks."

---

> > > > ### Comment · Reviewer_UsYi · 2023-08-16
> > > > **Not convinced that the framework + model are directly applicable to sparse graphs**
> > > >
> > > > > We respectfully disagree that our models “are not directly applicable to the sparse setting”.
> > > >
> > > > Perhaps one can sparsify the distance matrix without sacrificing completeness; as I said, this seems like an interesting research question on its own. You'd need to show this more formally, though. Which is why I believe the current theoretical framework is not directly applicable to the sparse setting.
> > > >
> > > > Next, regarding the models: I meant that these models operating on distance matrices (the input data structure is an n x n matrix) cannot be directly applied to sparse graphs where the input data structure is generally an edge index. The edge index could be converted to an n x n matrix, but this is obviously not nice for large n. Have I misunderstood?

---

> > > > > ### Author Response · Authors · 2023-08-17
> > > > > **Further clarification**
> > > > >
> > > > > We thank the prompt feedback from the reviewer, and we address the comments below.
> > > > >
> > > > > > Perhaps one can sparsify the distance matrix without sacrificing completeness; as I said, this seems like an interesting research question on its own. You'd need to show this more formally, though. Which is why I believe the current theoretical framework is not directly applicable to the sparse setting.
> > > > > >
> > > > >
> > > > > Thanks for the clarification on this point. We apologize for our misunderstanding. The sparse solutions we provide serve as the *trade-off* between completeness and efficiency. And our response/discussion about “the *potential* of distance matrices/DisGNNs to *maintain completeness when properly sparsified*” is currently a *preliminary/intuitive direction* for our models (or other models based on distance matrix). Further research is needed to explore this potential.
> > > > >
> > > > > > Next, regarding the models: I meant that these models operating on distance matrices (the input data structure is an n x n matrix) cannot be directly applied to sparse graphs where the input data structure is generally an edge index. The edge index could be converted to an n x n matrix, but this is obviously not nice for large n. Have I misunderstood?
> > > > > >
> > > > >
> > > > > Thanks for providing additional clarification regarding the concerns. However, we respectfully think there may be some misunderstanding. Though the current code implementation of DisGNNs we offer in our supplementary materials is in dense form (utilizing a dense n x n matrix as input), please note that our models are **not restricted to dense implementation**. Given the potential sparse solutions we provide (in previous response and Sec. 8 of our paper), by utilizing sparse data structure (i.e., sparse edge index) and “scatter” operations in torch (please see document of pytorch_scatter for more information), one can implement the sparse-version DisGNNs.
> > > > >
> > > > > We take 2-F-DisGNN for example. Given the embedding and interaction cutoff: 1) Tuples with lengthy diameters can be eliminated and the other tuples’ embeddings can be stored in a tensor $emb$ with shape (nnz, hidden_dim). 2) The interaction happening among 2-tuples can be indexed in a tensor $inter\\_idx$ in shape (nnz_inter, 2) (here, "2" corresponds to the 2-element tuples of 2-F-DisGNN in Eq. 3 in our paper), where each element records a possible pair of neighbors of some 2-tuple. 3) One can use $inter\\_idx$ to access the corresponding neighbors from $emb$, calculate the interaction results, and then aggregate/reduce them via "scatter" with an aggregation index tensor $agg$ with shape (nnz_inter) ranging from [0, nnz). This process is similar to the code implementation of DimeNet. Also please note that all the indices can be pre-calculated, making the overall process sparse and efficient.
> > > > >
> > > > > Thanks again for the careful comments. We hope our response can address the concerns, and if you have any other questions/concerns, please do not hesitate to let us know.

---

> > > > > > ### Comment · Reviewer_UsYi · 2023-08-20
> > > > > > **Thank you for the discussions**
> > > > > >
> > > > > > Thank you for the discussions, I have raised my score to an Accept.

---

### Author Rebuttal · Authors · 2023-08-10

We sincerely thank all the reviewers for their careful reading, and the thoughtful and detailed comments! The **figures** and **tables** referred to in separate responses are included in the following supplementary **PDF** file.

Best,

Authors.

---

### Decision · Program_Chairs · 2023-09-21

**Decision:**

Accept (poster)

**Comment:**

The paper studies the expressive power of GNNs on geometric graphs, i.e., graphs with Euclidean node embeddings. The paper discusses several failure cases and suggests novel GNN architecture that solves these failures. All reviewers as well as the AC appreciated the contributions in this paper and supported acceptance.